# Metabolic reprogramming of cancer cells by JMJD6-mediated pre-mRNA splicing associated with therapeutic response to splicing inhibitor

Carolyn M Jablonowski[1†], Waise Quarni[1†], Shivendra Singh[1], Haiyan Tan[2], Dhanushka Hewa Bostanthirige[1], Hongjian Jin[3], Jie Fang[1], Ti-Cheng Chang[3], David Finkelstein[3], Ji-Hoon Cho[2], Dongli Hu[1], Vishwajeeth Pagala[2], Sadie Miki Sakurada[4], Shondra M Pruett-Miller[4], Ruoning Wang[5], Andrew Murphy[1], Kevin Freeman[6], Junmin Peng[7], Andrew M Davidoff[1,8,9], Gang Wu[3], Jun Yang[1,8,9,10]*

[1]Department of Surgery, St Jude Children's Research Hospital, Memphis, United States; [2]Center for Proteomics and Metabolomics, St Jude Children's Research Hospital, Memphis, United States; [3]Center for Applied Bioinformatics, St Jude Children's Research Hospital, Memphis, United States; [4]Department of Cell and Molecular Biology, St Jude Children's Research Hospital, Memphis, United States; [5]Center for Childhood Cancer and Blood Disease, Abigail Wexner Research Institute, Nationwide Children's Hospital, Columbus, United States; [6]Genetics, Genomics & Informatics, The University of Tennessee Health Science Center (UTHSC), Memphis, United States; [7]Department of Structural Biology, St Jude Children's Research Hospital, Memphis, United States; [8]St Jude Graduate School of Biomedical Sciences, St Jude Children's Research Hospital, Memphis, United States; [9]Department of Pathology and Laboratory Medicine, College of Medicine, The University of Tennessee Health Science Center, Memphis, United States; [10]College of Graduate Health Sciences, University of Tennessee Health Science Center, Memphis, United States

*For correspondence: Jun.Yang2@stjude.org

†These authors contributed equally to this work

**Abstract** Dysregulated pre-mRNA splicing and metabolism are two hallmarks of MYC-driven cancers. Pharmacological inhibition of both processes has been extensively investigated as potential therapeutic avenues in preclinical and clinical studies. However, how pre-mRNA splicing and metabolism are orchestrated in response to oncogenic stress and therapies is poorly understood. Here, we demonstrate that jumonji domain containing 6, arginine demethylase, and lysine hydroxylase, JMJD6, acts as a hub connecting splicing and metabolism in MYC-driven human neuroblastoma. JMJD6 cooperates with MYC in cellular transformation of murine neural crest cells by physically interacting with RNA binding proteins involved in pre-mRNA splicing and protein homeostasis. Notably, JMJD6 controls the alternative splicing of two isoforms of glutaminase (GLS), namely kidney-type glutaminase (KGA) and glutaminase C (GAC), which are rate-limiting enzymes of glutaminolysis in the central carbon metabolism in neuroblastoma. Further, we show that JMJD6 is correlated with the anti-cancer activity of indisulam, a 'molecular glue' that degrades splicing factor RBM39, which complexes with JMJD6. The indisulam-mediated cancer cell killing is at least partly dependent on the glutamine-related metabolic pathway mediated by JMJD6. Our findings reveal a cancer-promoting metabolic program is associated with alternative pre-mRNA splicing through JMJD6, providing a rationale to target JMJD6 as a therapeutic avenue for treating MYC-driven cancers.

## eLife assessment

This **important** study reports on key characteristics of MYC-driven cancers: dysregulated pre-mRNA splicing and altered metabolism, with the data being overall **solid**. The manuscript should be of broad interest to cancer biologists due to its therapeutic implications.

## Introduction

Metabolic reprogramming is a hallmark of cancer (*Hanahan and Weinberg, 2011*; *Pavlova and Thompson, 2016*; *Vazquez et al., 2016*), which allows rapidly proliferating tumor cells to acquire nutrients to meet their bioenergetic, biosynthetic, and redox demands (*DeBerardinis and Chandel, 2016*). One of the primary driving forces in reprogramming cancer cell metabolism is the deregulated MYC family proto-oncogenes (*MYC, MYCN,* and *MYCL*) (*Kalkat et al., 2017*), which are known to encode master transcriptional factors that regulate metabolic gene expression. MYC coordinates nutrient acquisition to produce ATP and key cellular building blocks that increase cell mass and promote DNA replication and cell division (*Stine et al., 2015*). The increase in total RNA and protein synthesis by overactive MYC signaling leads to dysregulation of macromolecular processing machineries including the spliceosome (*Hsu et al., 2015*), and consequently pre-mRNA splicing (*Hirsch et al., 2015*; *Koh et al., 2015*; *Phillips et al., 2020*), another hallmark of MYC-driven cancers (*Hsu et al., 2015*; *Koh et al., 2015*; *Phillips et al., 2020*; *Anczuków and Krainer, 2015*; *Zhang et al., 2016*), for the purpose of cellular stress adaptation. *MYCN* amplification is one of the most important biological features of high-risk neuroblastoma (*Gustafson and Weiss, 2010*). Transgenic mouse and zebrafish models have demonstrated that MYCN is a neuroblastoma driver (*Weiss et al., 1997*; *Zhu et al., 2012*). In tumors without *MYCN* amplification, *MYC* is overexpressed, further indicating that neuroblastoma is a MYC-driven cancer. The metabolic dependency of neuroblastoma has been widely studied by us and others (*Tao et al., 2022*; *Khan et al., 2020*; *Gamble et al., 2019*; *Xia et al., 2019*; *Wang et al., 2018*; *Bansal et al., 2022*; *Olsen et al., 2022*; *Alborzinia et al., 2022*). A larger number of splicing changes have also been noticed in high-stage neuroblastomas (*Guo et al., 2011*; *Shi et al., 2021*). Splicing alterations lead to a spliceosomal vulnerability that provides a new opportunity to develop transformative therapies by disrupting aberrant pre-mRNA splicing (*Hsu et al., 2015*; *Koh et al., 2015*; *Phillips et al., 2020*; *Anczuków and Krainer, 2015*; *Zhang et al., 2016*). We and others have shown that targeting the splicing factor RBM39 by indisulam, a 'molecular glue' that bridges RBM39 to E3 ubiquitin ligase DCAF15 for proteasomal degradation, achieved significant anti-tumor activity in neuroblastoma models (*Singh et al., 2021*; *Nijhuis et al., 2022*). Disruption of spliceosome by pladienolide B also resulted in significant anti-tumor effect in neuroblastoma models (*Shi et al., 2021*). However, how the dysregulated pre-mRNA splicing machinery and metabolism are orchestrated in MYC-driven neuroblastoma has not been well elucidated. Whether metabolism modulates the anti-cancer effect of splicing inhibition remains to be answered.

Next-generation sequencing studies have revealed only a few recurrent somatic mutations in neuroblastoma at the time of diagnosis (*Pugh et al., 2013*; *Molenaar et al., 2012*). However, copy number alterations of chromosomal segments such as 17q gain, 1p36 or 11q23 loss frequently occur in high-risk neuroblastoma. While attempts to understand the functions of individual genes in these chromosomal segments have been reported (i.e. *BIRC5* [*Hagenbuchner et al., 2016*], *PHB* [*MacArthur et al., 2019*], *PPM1D* [*Milosevic et al., 2021*], *TRIM37* [*Meitinger et al., 2020*] in 17q, *ARID1A* [*García-López et al., 2020*], *CAMTA1* [*Henrich et al., 2011*], *CASZ1* [*Liu et al., 2011*], *CHD5* [*Laut et al., 2022*; *Higashi et al., 2015*; *Fujita et al., 2008*], *KIF1B* [*Fell et al., 2017*; *Li et al., 2016*; *Chen et al., 2014*], *MIR34A* [*Cole et al., 2008*], *RUNX3* [*Yu et al., 2014*] in 1p36), the biological consequences of these genetic events in MYC-driven tumors still remain largely unknown. Gain of 17q is the most frequent genetic event in high-risk neuroblastoma and is associated with *MYCN* amplification (*Bown et al., 1999*). In addition, in the transgenic *MYCN* mouse model of neuroblastoma, the chromosomal locus syntenic to human 17q is partially amplified (*Althoff et al., 2015*), indicating that chromosome 17q is needed for MYC-mediated tumorigenesis.

JMJD6 is a JmjC domain-containing nuclear protein with iron- and 2-oxoglutarate-dependent dioxygenase activity (*Böttger et al., 2015*), whose coding gene is located on chromosome 17q25. While the histone arginine demethylase activity of JMJD6 that catalyzes demethylation of H4R3me1/me2 is controversial (*Chang et al., 2007*), JMJD6 is a lysyl-5-hydroxylase that catalyzes 5-hydroxylation

on specific lysine residues of target proteins (*Webby et al., 2009*). JMJD6 has pleiotropic functions in normal physiology and in cancer (*Kwok et al., 2017*; *Vangimalla et al., 2017*; *Zhou et al., 2022*; *Paschalis et al., 2021*). We previously found that JMJD6 is essential for the survival of neuroblastoma cells (including *MYCN*-amplified and *MYC*-overexpressed cells) (*Yang et al., 2017*), which was further validated by an independent study (*Wong et al., 2019*), indicating that neuroblastoma has JMJD6 dependency. However, the exact mechanism of JMJD6 in MYC-driven cancers remains elusive. One study has shown that JMJD6 and BRD4 co-bind at antipause enhancers, regulating promoter-proximal pause release of a large subset of transcription units (*Liu et al., 2013*). By harnessing a similar mechanism, JMJD6 promotes cell survival of glioblastoma in vivo (*Miller et al., 2017*). These findings are particularly interesting because BRD4 occupies exceptionally large super-enhancers associated with genes, including *MYC* and *MYCN* (*Lovén et al., 2013*; *Chapuy et al., 2013*; *Puissant et al., 2013*), and the expression of those enhancers can be disrupted by BRD4 inhibitors, which have a potent anti-tumor effect (*Lovén et al., 2013*; *Chapuy et al., 2013*; *Puissant et al., 2013*; *Wyce et al., 2013*). Here, we show a new mechanism by which JMJD6 promotes tumorigenesis mediated by the MYC oncogene in that JMJD6 interacts with a subset of RNA binding proteins including RBM39 in neuroblastoma cells and regulates the alternative splicing of metabolic genes that are involved in mitochondrial metabolism. 'Glutamine addiction' is one key feature of MYC-driven tumors. Glutaminase (GLS) is the enzyme responsible for conversion of glutamine to glutamate in the process of glutaminolysis to feed the tricarboxylic acid (TCA) cycle and has two splice isoforms, GAC (glutaminase C) and KGA (kidney-type glutaminase). We show that JMJD6 controls the alternative splicing of KGA and GAC, and, consequently, impacts the central carbon metabolism in neuroblastoma. Further, we show that JMJD6 is correlated with the anti-cancer activity of indisulam, a 'molecular glue' that degrades the splicing factor RBM39. The indisulam-mediated cancer cell killing is at least partly dependent on the glutamine-related metabolic pathway mediated by JMJD6. Our findings demonstrate a new mechanism by which JMJD6 coordinates metabolic programs and alternative pre-mRNA splicing, providing a rationale to target JMJD6 as a therapeutic target for MYC-driven cancers.

## Results

### The essential genes for neuroblastoma cell survival on chromosome 17q target pre-mRNA splicing and metabolism

An incomplete understanding of the biological consequences of chromosome 17q gain remains a barrier to the understanding of high-risk neuroblastoma. 1132 genes are located on 17q (*Figure 1— source data 1*). We surmised that some of the 17q genes are particularly important for neuroblastoma cell survival. Analysis of the cancer dependency genes in neuroblastoma cell lines screened with the Avana sgRNA library (*Meyers et al., 2017*) revealed that 114 were essential to neuroblastoma (mean score $<-0.4$) (*Figure 1A*, *Figure 1—source data 1*). Protein interaction network analysis followed by functional annotation revealed that proteins encoded by these 114 essential genes formed distinct but interconnected modules including RNA splicing (i.e. *SRSF2*, *DDX5*, *DDX42*, *DHX8*), mitochondrial metabolism (i.e. *NDUFA8*, *COX11*, *SLC25A10*, *SLC35B1*), protein homeostasis (i.e. *UBE2O*, *PSMB3*, *PSMC5*), DNA repair (i.e. *BRIP1*, *BRCA1*, *RAD51C*, *RAD51D*) and transcriptional regulation (i.e. *PHF12*, *CBX1*, *SMARCE1*, *MED1*), as well as endocytosis (i.e. *CHMP6*, *CTLC*, *EPN3*, *HGS*, *SNF8*, *VPS25*) (*Figure 1B*). Children aged ≥18 months with metastatic disease and patients with MYCN amplification tumors are classified as high-risk, which requires a multimodal therapy including induction chemotherapy, surgical resection of primary disease, consolidation with high-dose chemotherapy and stem cell rescue, radiotherapy, and post-consolidation treatment with cis-retinoic acid and immunotherapy (*Morgenstern et al., 2019*). Using these 114 genes as a signature, we found that 81 of them were highly expressed in high-risk neuroblastomas, which were enriched with MYCN amplification (*Figure 1C*). Correspondingly, neuroblastomas with high expression levels of this gene signature were associated with a poorer event-free (time from treatment until the cancer progresses) and overall survival (time from treatment to death) of patients in two large clinic cohorts (*Figure 1D and E*, *Figure 1—figure supplement 1*). Interestingly, in low-risk neuroblastoma patients, high expression of the 114 essential genes was associated with poor event-free and overall survival, while no difference was observed in high-risk patients (*Figure 1—figure supplement 2*). These data demonstrate that

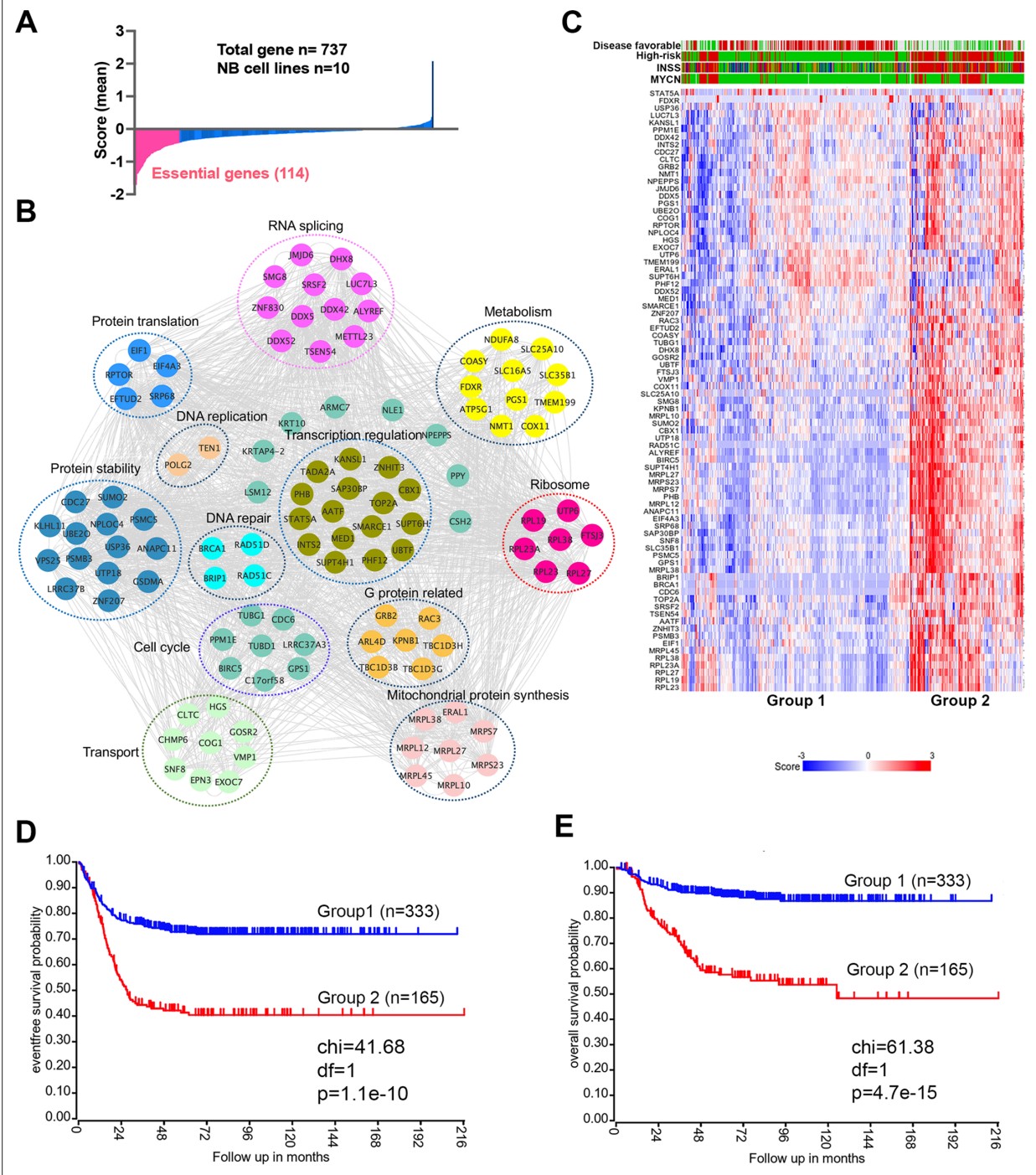

**Figure 1.** 17q contains neuroblastoma dependency genes. (**A**) CRISPR score for 17q genes in 10 neuroblastoma cell lines. Score <−0.4 is defined as neuroblastoma dependency genes. Data are derived from Avana sgRNA library screening (***Meyers et al., 2017***). (**B**) STRING protein interaction network showing 17q essential genes with various biological functions. (**C**) Heatmap by K-means clustering analysis showing 17q essential genes are highly expressed in high-risk neuroblastomas based on RNA-seq data (SEQC dataset). (**D**) Kaplan-Meier survival curve showing 17q essential gene signature is correlated with worse event-free survival (SEQC dataset). (**E**) Kaplan-Meier survival curve showing 17q essential gene signature is correlated with worse overall survival (SEQC dataset).

The online version of this article includes the following source data and figure supplement(s) for figure 1:

**Source data 1.** 17q gene list.

**Figure supplement 1.** High expression of 17q essential genes is associated with worse event-free and overall survival.

*Figure 1 continued on next page*

*Figure 1 continued*

**Figure supplement 2.** High expression of 17q essential genes is associated with worse event-free and overall survival in low-risk neuroblastomas but not high-risk neuroblastomas.

17q genes are involved in essential biological processes and highly expressed in high-risk neuroblastomas. Nevertheless, the 114 essential genes cannot further stratify the high-risk patients.

## JMJD6 is required for neuroblastoma growth

*JMJD6* was among these 114 essential genes. To understand the role of JMJD6, we examined the genetic features of *JMJD6* in neuroblastoma and other types of cancers. Among the genes encoding JmjC domain containing proteins, *JMJD6* was the only one that was frequently amplified in neuroblastoma (*Figure 2A*). High *JMJD6* expression was associated with poor event-free outcome, as shown by Kaplan-Meier analysis (*Figure 2B*). Further analysis of JMJD6 expression in low-risk and high-risk patients showed that high levels of JMJD6 expression was associated with poor event-free and overall survival in both low-risk and high-risk patients (*Figure 2—figure supplement 1*), indicating that JMJD6 is high-risk factor regardless of disease status. To examine whether *JMJD6* amplification is limited to specific tumor types, we explored genomic data from different cancers using the cBioportal program (*Cerami et al., 2012*). *JMJD6* was amplified across multiple types of adult cancers such as breast and liver cancer (*Figure 2—figure supplement 2A*), and correlated with worse relapse-free survival (*Figure 2—figure supplement 2B*). We further compared the RNA-seq expression of *JMJD6* in 2337 samples across over 20 pediatric cancer subtypes and found that *JMJD6* showed the highest expression levels in neuroblastoma (*Figure 2—figure supplement 2C*), suggesting that JMJD6 might be particularly important in neuroblastoma. We validated this hypothesis using shRNA knockdown of JMJD6 in *MYCN*-amplified cells (BE2C, SIMA, KELLY, IMR32) and non-*MYCN*-amplified cells (SK-N-AS and CHLA20). The results showed that loss of JMJD6 greatly reduced the colony numbers in all tested cell lines (*Figure 2C* and Materials and methods), demonstrating that JMJD6 is essential to neuroblastoma cells regardless of *MYCN* amplification. Neuroblastic tumors comprise a histological spectrum that ranges from less-differentiated neuroblastoma to well-differentiated ganglioneuroma. The extent of differentiation in the tumor cells is correlated with prognostic significance (*Brodeur and Bagatell, 2014*). We noticed that the loss of JMJD6 led to neurite outgrowth (*Figure 2—figure supplement 2D*), a unique structure of neuroblastoma cells differentiating in vitro. This morphological change suggests that JMJD6 is required to regulate cellular differentiation. Lastly, we validated that JMJD6 is essential to neuroblastoma growth in *MYCN*-amplified (BE2C) and MYC-overexpressed (SK-N-AS) xenograft models (*Figure 2D and E*). Taken together, these data demonstrate that loss of JMJD6 function impedes neuroblastoma cell survival and tumor growth.

## JMJD6 promotes MYC-mediated cellular transformation

Next, we investigated whether gain of function of JMJD6 could facilitate MYC-mediated oncogenic transformation. To test this, we used an NIH3T3 transformation assay that provides a straightforward method to assess the transforming potential of an oncogene, which may lead to morphological transformation and loss of contact inhibition, a typical feature of cellular transformation. Like the *GFP* control (*Figure 2—figure supplement 3A*), we found that NIH3T3 cells with overexpressed *JMJD6* stopped proliferation after being confluent (*Figure 2—figure supplement 3B*), indicating *JMJD6* alone is unable to transform NIH3T3 cells. However, overexpression of *MYCN* induced foci formation with enhanced cell death (*Figure 2—figure supplement 3C*). Importantly, NIH3T3 cells with overexpressed *MYCN* and *JMJD6* lost contact inhibition, accompanied with morphological change, and formed larger foci (*Figure 2—figure supplement 3D*), indicating that *JMJD6* enhances *MYCN* activity to transform NIH3T3 cells. Interestingly, *MYCN* alone also reprogrammed metabolism of NIH3T3 cells as shown by the color change of the media, which was largely rescued by co-expression of *JMJD6* (*Figure 2—figure supplement 3E*), suggesting that cells with enhanced lactate production by MYCN were directed to oxidative phosphorylation by JMJD6. It is believed that the cell of origin of neuroblastoma is the progeny of neural crest cells (*Kameneva et al., 2021*; *Jansky et al., 2021*). We therefore tested the role of JMJD6 in MYC-mediated transformation using JoMa1 (*Maurer et al., 2007*), a cell line derived from murine neural crest, by transducing *GFP*, *JMJD6*, *MYCN,* and *JMJD6/MYCN* (*Figure 2F*). While *JMJD6* showed no difference from *GFP* control in regulating cell proliferation,

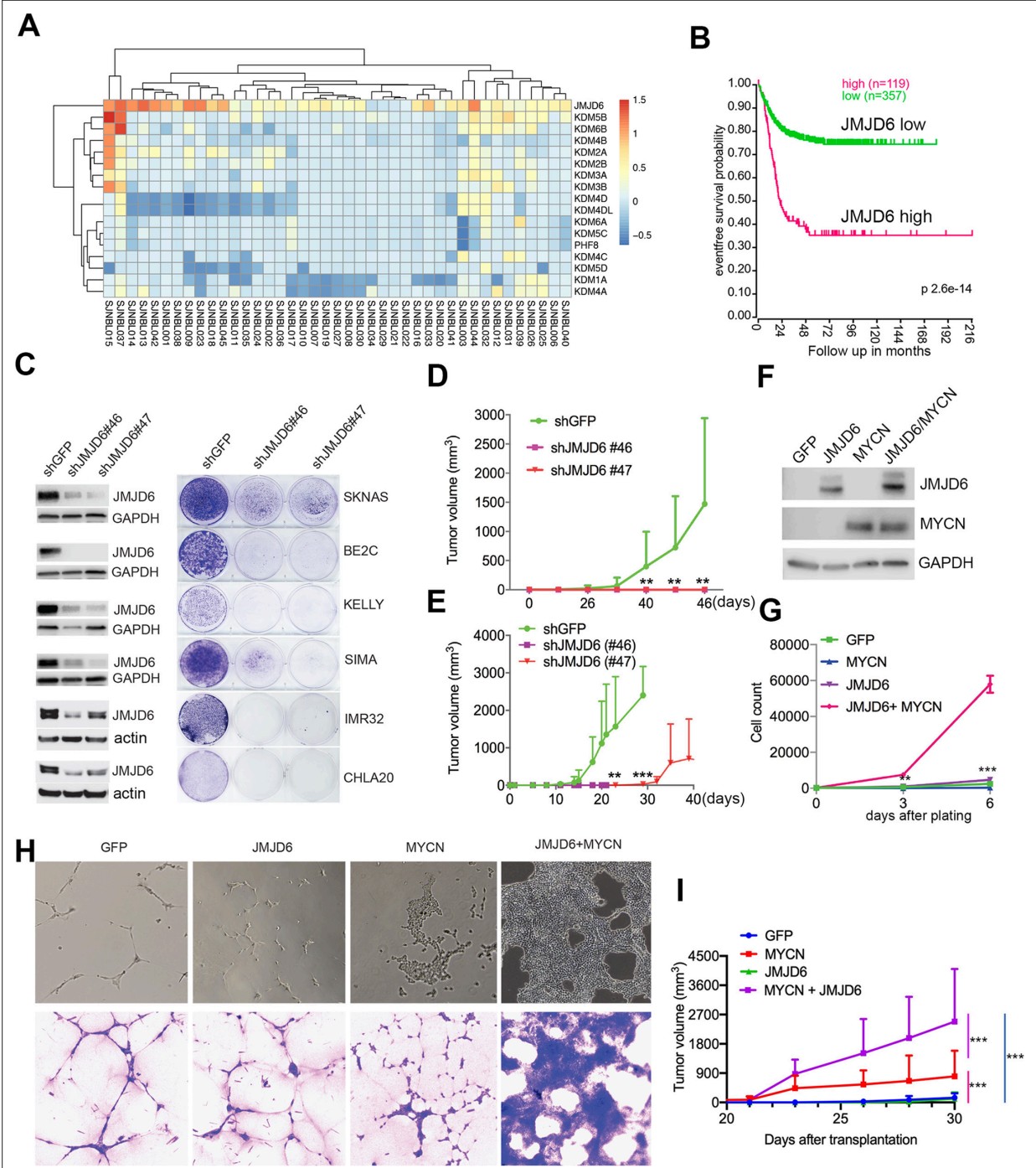

**Figure 2.** JMJD6 is required for neuroblastoma growth and facilitates MYC-mediated cellular transformation. (**A**) Copy number of genes encoding JmjC domain proteins in St Jude neuroblastoma cohort (https://platform.stjude.cloud). (**B**) Kaplan-Meier survival curve showing high JMJD6 is correlated with worse event-free survival (SEQC RNA-seq dataset). (**C**) Crystal violet showing the colony staining on day 7 after JMJD6 shRNA knockdown in neuroblastoma cell lines validated by western blot (harvested at 72 hr). n=single experiment. (**D**) Xenograft tumor growth of BE2C (right) models with lentiviral JMJD6 shRNA knockdown. p-Value calculated by multiple unpaired t-test across each row. n=5 per group. ***p<0.001, **p<0.01. (**E**) Xenograft tumor growth of SK-N-AS models with lentiviral JMJD6 shRNA knockdown. p-value calculated by multiple unpaired t-test across each row. ***p<0.001, **p<0.01. (**F**) Western blot validating the expression of retroviral-based MYCN and JMJD6 in JoMa1 cells. (**G**) Cell proliferation of JoMa1 cells transduced with indicated constructs expressing GFP, JMJD6, MYCN, JMJD6+MYCN. (**H**) Colony formation of JoMa1 cells transduced with indicated constructs, GFP, JMJD6, MYCN, JMJD6+MYCN. Top panel showing photos taken under light microscope. Bottom panel showing cell colonies stained with crystal violet. (**I**) Xenograft tumor growth of JoMa1 cells transduced with indicated constructs, GFP, JMJD6, MYCN, JMJD6+MYCN. n=5 per group. p-Value calculated by multiple unpaired t-test across each row. ***p<0.001, **p<0.01. Data are Mean ± SEM.

*Figure 2 continued on next page*

**Figure supplement 2.** JMJD6 expression in cancers and effect on neuroblastoma differentiation.

**Figure supplement 3.** JMJD6 coordinates with MYC to transform.

*MYCN* slightly enhanced cell proliferation (*Figure 2G*). However, the combination of *JMJD6* and *MYCN* remarkably increased cell proliferation, mirrored by the colony formation assay which showed *JMJD6/MYCN*-induced rapid growth of colonies with distinct transformation morphology (*Figure 2H*). Implantation of each group into immune-deficient mice led to tumor development of *MYCN* and *JMJD6/MYCN* groups (*Figure 2I*). However, *JMJD6/MYCN* group tumors appeared to grow faster than the *MYCN* alone tumors. Taken together, these data indicate that JMJD6 enhances the MYC-mediated transformation, demonstrating the oncogenic role of JMJD6.

## JMJD6 regulates pathways engaged in pre-mRNA splicing and mitochondrial biogenesis in neuroblastoma

We surmised that loss of function of genes in the same functional module/pathway may induce similar effects across different cell lineages, which in turn corroborates the hypothesis that JMJD6 is a player in that signaling pathway. To test this hypothesis, we analyzed the dependency correlation of *JMJD6* knockout and other genes (defined as co-dependency genes if they are positively correlated), by using the DepMap data (https://depmap.org) that includes genome-wide knockout in 1027 cell lines of more than 20 cancer types (*Figure 3A–D*; *Figure 3—source data 1*), followed by pathway enrichment. The data showed that JMJD6 co-dependency genes were significantly and positively correlated with spliceosome/mRNA splicing (i.e. RBM39, SF3B1), ubiquitin-mediated proteolysis and endocytosis and a number of 17q25 genes (*Figure 3A and C*), which mirrored the pathway network of 17q essential genes in neuroblastoma (*Figure 1B*). JMJD6 knockout was negatively correlated with the knockout of genes housed at chromosome 1p, which is frequently deleted in high-risk neuroblastoma, and oxidative phosphorylation as well as protein translation (*Figure 3B and D*).

BRD4 is known to regulate MYC expression (*Puissant et al., 2013*; *Delmore et al., 2011*). Previous studies have shown that JMJD6 and BRD4 interact to regulate gene transcription (*Wong et al., 2019*; *Miller et al., 2017*), suggesting that JMJD6 might directly modulate MYC expression. To assess this possibility, we knocked down *JMJD6* in neuroblastoma cell lines BE2(C) and SK-N-AS, which express *MYCN* and *MYC*, respectively, for RNA-seq analysis. The sequencing data were analyzed for differential gene expression (*Figure 3—figure supplement 3*). Interestingly, loss of JMJD6 showed minimal impact on expression of *MYCN* in BE2C cells or *MYC* in SK-N-AS cells (*Figure 3—figure supplement 1A*), and western blot analysis did not show alteration of MYCN expression although MYC protein was slightly downregulated by loss of JMJD6 (*Figure 3—figure supplement 1B*). However, BRD4 inhibitors drastically inhibited both MYCN and MYC expression in neuroblastoma cells (*Lovén et al., 2013*; *Chapuy et al., 2013*; *Puissant et al., 2013*), suggesting that JMJD6 inhibition has a distinct effect from the BRD4 inhibition. Gene set enrichment analysis (GSEA) for pathway engagement for the genes commonly downregulated or upregulated in both cell lines revealed that loss of JMJD6 most significantly repressed the expression of genes involved in pre-mRNA splicing, histones, and cell cycle G1/S checkpoint (*Figure 3E*), and enhanced the pathways involved in mitochondrial functions and heat shock response (*Figure 3E*). Interestingly, the genes transcribed from the mitochondrial genome were elevated in both cell lines (*Figure 3—figure supplement 1C*), suggesting that JMJD6 directly or indirectly regulates the transcription of mitochondrial genome. These data are consistent with the co-dependency pathways of JMJD6 (*Figure 3A–D*). Depletion of JMJD6 in both cell lines led to the downregulation of MYC signaling pathways (*Figure 3—figure supplement 2A*), although ranked behind the pathways of splicing and metabolism, suggesting that the MYC pathways are not primarily regulated by JMJD6. These data indicate that JMJD6 does not regulate the gene expression of the MYC family of transcription factors but might indirectly regulate the MYC pathway. Nevertheless, loss of JMJD6 induced an induction of gene signatures related to axon, neuron projection, and Schwann cell differentiation (*Figure 3—figure supplement 2B*), which is consistent with the induction

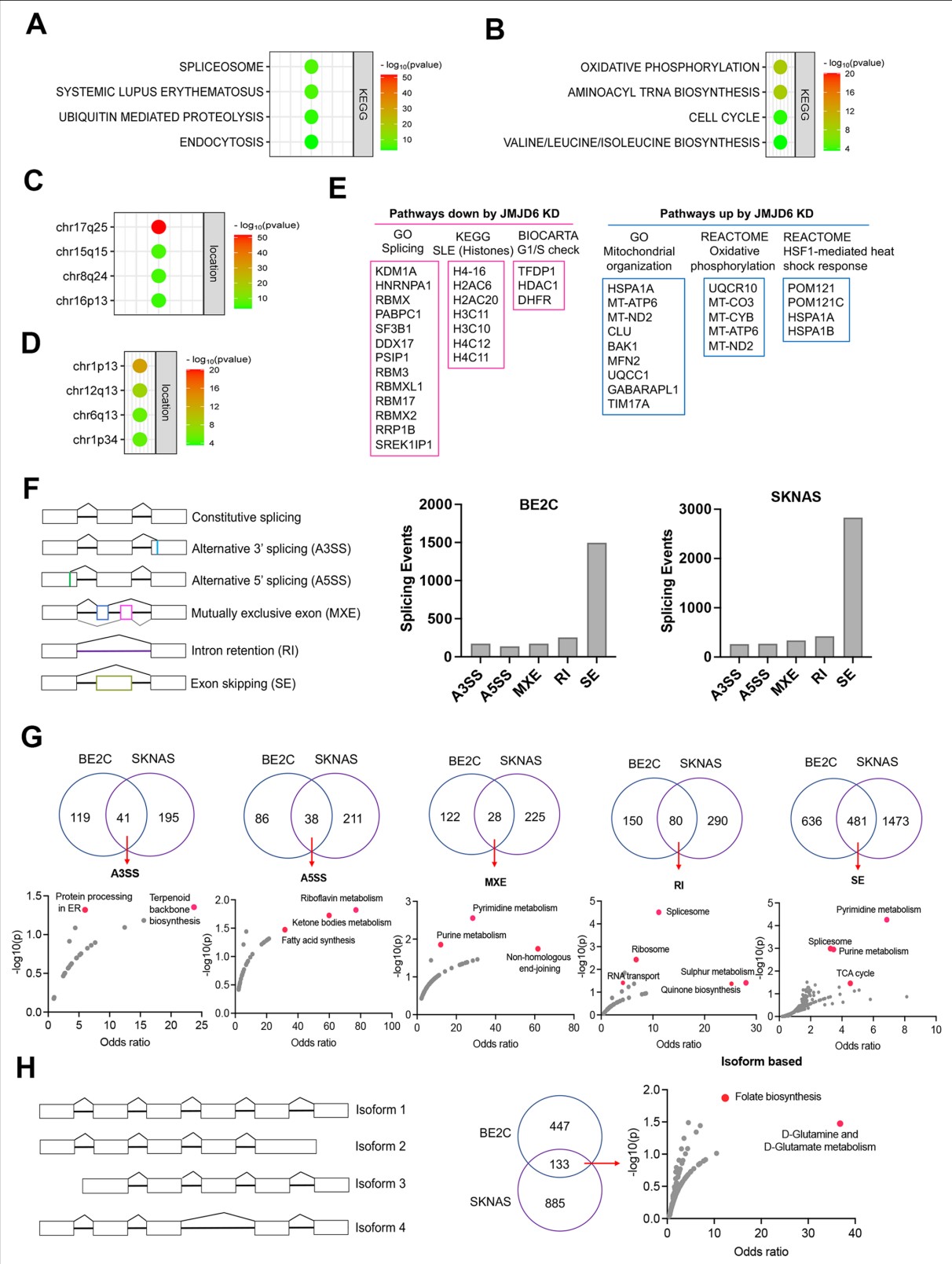

**Figure 3.** JMJD6 regulates pre-mRNA splicing of genes involved in metabolism. (**A**) Pathway enrichment for JMJD6 co-dependency genes whose knockout exhibits similar phenotype with JMJD6 knockout based on re-analysis of DepMap data. (**B**) Pathway enrichment for genes whose knockout exhibits opposite phenotype with JMJD6 knockout based on re-analysis of DepMap data. (**C**) Chromosomal location enrichment for JMJD6 co-dependency genes whose knockout exhibits similar phenotype with JMJD6 knockout based on re-analysis of DepMap data. (**D**) Chromosomal location

*Figure 3 continued on next page*

Figure 3 continued

enrichment for genes whose knockout exhibits opposite phenotype with JMJD6 knockout based on re-analysis of DepMap data. (**E**) Pathway analysis for genes downregulated and upregulated (cutoff, log2FC = 1.7) by JMJD6 knockdown commonly shared in SK-NAS and BE2C cells. (**F**) Alternative splicing events altered by JMJD6 knockdown in BE2C and SK-N-AS cells. (**G**) Pathway enrichment for the genes with each splicing event commonly altered in BE2C and SK-N-AS cells after JMJD6 knockdown. (**H**) Isoform identification based on splicing events in BE2C and SK-N-AS cells, followed by pathway enrichment for commonly shared alterations in both cell lines.

The online version of this article includes the following source data and figure supplement(s) for figure 3:

**Source data 1.** Differential gene expression after JMJD6 knockdown in BE2C and SKNAS cells.

**Source data 2.** Pathways affected by JMJD6 knockdown in BE2C and SKNAS cells.

**Figure supplement 1.** JMJD6 knockdown does not affect MYC expression but upregulates mitochondrial gene expression.

**Figure supplement 2.** Pathways affected by JMJD6 knockdown.

**Figure supplement 3.** JMJD6 regulates pre-mRNA splicing.

---

of neurite outgrowth observed in BE2C cells after JMJD6 knockdown (*Figure 2—figure supplement 2D*), indicative of neuroblastoma cell differentiation.

To verify if JMJD6 regulates pre-mRNA splicing, we analyzed the RNA-seq using two algorithms (*Wu et al., 2018*). The first one is event-based analysis to identify the altered exon splicing (*Figure 3F*, *Figure 3—source data 1*). We found that knockdown of JMJD6 dominantly affects the exon skipping although other splicing events were also altered, albeit with a much smaller number (*Figure 3F*). Pathway analysis of common events in both BE2C and SK-N-AS cells demonstrated that genes involved in metabolism and splicing are most significantly affected by loss of function of JMJD6 (*Figure 3G*). Using the second algorithm of RNA splicing analysis previously developed (*Figure 3H*, *Figure 3—source data 1*) that allows discovery of new isoforms of genes generated through alternative splicing (*Wu et al., 2018*), we identified 580 genes in BE2C cells and 1018 genes in SK-N-AS cells undergoing alternative splicing after JMJD6 knockdown, 133 of which were shared by both (*Figure 3H*, *Figure 3—figure supplement 3*, *Figure 3H—source data 1*). The alternatively spliced genes were involved in a variety of pathways (*Figure 3—figure supplement 3*). Among the 133 commonly alternatively spliced genes, KEGG pathway enrichment analysis showed that metabolic pathway genes were the only ones significantly enriched (*Figure 3H*, *Figure 3—source data 1*), most of which are involved in mitochondrial bioenergetics and folate metabolism (*Figure 3H*). Collectively, these data demonstrate that JMJD6 regulates RNA splicing of genes engaged in mitochondrial metabolism, being one of the key mediators of the 17q locus activity in neuroblastoma.

## JMJD6 regulates alternative splicing of glutaminolysis gene GLS

Overactive MYC signaling leads to altered macromolecular processing machineries in response to an increase in total RNA and protein synthesis (*Hsu et al., 2015*). MYC is also a master regulator of cancer metabolism involved in ribosomal and mitochondrial biogenesis, glucose and glutamine metabolism, and lipid synthesis, leading to the acquisition of bioenergetic substrates enabling the cancer cell to grow and proliferate (*Dang, 2013*; *Miller et al., 2012*; *Gordan et al., 2007*). 'Glutamine addiction' is one feature of MYC-driven cancer (*Wise and Thompson, 2010*). The pre-mRNA splicing altered by JMJD6 knockdown included *GLS*, the key enzyme of glutaminolysis, prompting us to investigate the *GLS* splicing mediated by JMJD6 knockdown. GLS is known to catalyze the conversion of glutamine to glutamate, and is alternatively spliced to form two isoforms, GAC and KGA (*Porter et al., 2002*), with different cellular localization and catalytic capacities (*Cassago et al., 2012*). The GAC isoform is more frequently upregulated in cancer cells than KGA (*Wang et al., 2010*), and has been shown to be regulated by MYC (*Gao et al., 2009*; *Wise et al., 2008*; *Yuneva et al., 2007*), leading to a 'glutamine addiction' phenotype in MYC-driven tumors (*Wise and Thompson, 2010*). We found that loss of JMJD6 led to a splicing switch from the GAC isoform (with exons 1–15) to the KGA isoform (with exons 1–14 and 16–19) (*Figure 4A*, *Figure 4—figure supplement 1*), which was further confirmed by isoform-specific real-time (RT) polymerase chain reaction (PCR) (*Figure 4B*). We then investigated the expression of GAC/KGA at the protein levels after JMJD6 knockdown. Western blot showed that the KGA isoform was increased after JMJD6 knockdown in all three tested cell lines, MYC overexpressed SK-N-AS, BE2C and SIMA with MYCN amplification (*Figure 4C*). Then, we further validated the JMJD6 effect on GLS isoform expression by using a luciferase reporter that indicates

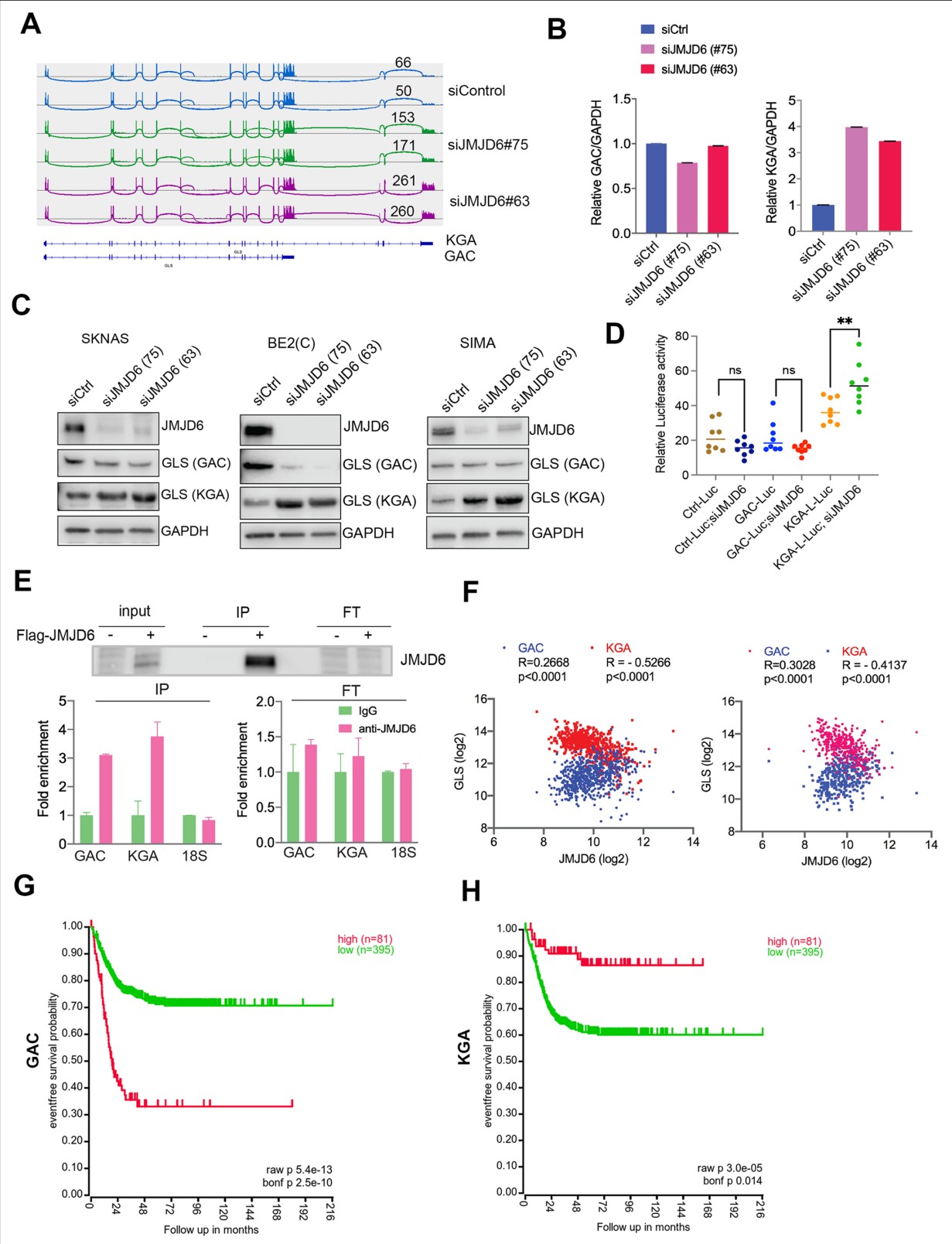

**Figure 4.** JMJD6 regulates alternative splicing of glutaminolysis gene, GLS. (**A**) Sashimi plot showing the alternative splicing of *GLS* after JMJD6 knockdown in BE2C cells in duplicates. The number indicates the RNA-seq read counts of exon junction. (**B**) Real-time (RT)-polymerase chain reaction (PCR) assessing the relative expression of *GAC* and *KGA* isoforms after JMJD6 knockdown in BE2C cells in triplicates. (**C**) Western blot showing the expression of GAC and KGA isoforms in SK-N-AS, BE2C, SIMA after JMJD6 knockdown for 72 hr. (**D**) KGA- and GAC-specific reporter analysis showing

*Figure 4 continued on next page*

*Figure 4 continued*

only KGA-driven luciferase activity is significantly upregulated by JMJD6 knockdown. (**E**) RNA immunoprecipitation showing JMJD6 interaction with *GLS* RNA (n=single experiment). Top panel shows the western blot analysis of FLAG-tagged JMJD6 in input, immunoprecipitation (IP), and flowthrough (FT) fractions. Bottom panel shows RT-PCR (n=3) analysis of enrichment of GAC/KGA bound by JMJD6 in IP and FT fractions. (**F**) Spearman correlation analysis of *JMJD6* and *GAC/KGA* expression levels in two neuroblastoma cohorts GSE45547 (left) and GSE120572 (right). (**G**) Kaplan-Meier curve showing the association of high or low *GAC* expression levels with event-free survival in a cohort of neuroblastoma (GSE45547). Expression cutoff = 3971 for GAC. (**H**) Kaplan-Meier curve showing the association of high or low *KGA* expression levels with event-free survival in a cohort of neuroblastoma (GSE45547). Expression cutoff = 7253 for KGA. Data are Mean ± SEM.

The online version of this article includes the following figure supplement(s) for figure 4:

**Figure supplement 1.** JMJD6 regulates the alternative splicing of GLS.

the isoforms of GAC and KGA. Indeed, JMJD6 knockdown significantly increased the expression of the KGA reporter (*Figure 4D*). RNA immunoprecipitation showed that JMJD6 bound to *GLS* RNA (*Figure 4E*), suggesting that JMJD6 may directly regulate the splicing of *GLS*. We reasoned that if the regulation of *GLS* splicing by JMJD6 was a bone fide mechanism, the expression levels of *JMJD6* would correlate with the levels of *GAC/KGA* in tumors. Indeed, *JMJD6* was positively correlated with *GAC* and negatively correlated with *KGA* in two independent neuroblastoma cohorts (*Figure 4F*), supporting the hypothesis that JMJD6 is required to maintain the high ratio of *GAC/KGA* in cancer cells by controlling their alternative splicing. Clinical relevance of *GAC* and *KGA* in neuroblastoma was evidenced by the findings that high *GAC* was associated with a worse event-free survival and high *KGA* was associated with a better event-free survival (*Figure 4G and H*), suggesting that the *GAC/KGA* ratio may play a role in cancer progression.

## GAC and KGA are both important for cell survival

To understand if GAC and KGA play distinct roles in neuroblastoma cells, we transduced GAC or KGA into BE2C and SKNAS cells. However, introduction of either GAC or KGA in BE2C cells or SKNAS cells promoted colony formation (*Figure 5A–D*), indicating that enhanced glutaminolysis by either GAC or KGA overexpression is pro-proliferative. Interestingly, RNA-seq analysis revealed that GAC and KGA share common targets but also have distinct targets in both BE2C and SKNAS cells (*Figure 5—figure supplement 1A, B*). KEGG pathway analysis of the commonly upregulated genes by GAC and KGA showed that both GAC and KGA promoted the PI3K-AKT and MAPK pathways in both cell lines (*Figure 5E and F*). However, for the genes commonly downregulated by GAC and KGA, KEGG pathway showed that calcium signaling was significantly downregulated in BE2C cells while steroid biosynthesis was significantly downregulated in SKNAS cells (*Figure 5E and F*). Then, we examined GAC- and KGA-specific effects in both cell lines. In BE2C cells, GAC expression promoted Hedgehog, Hippo, and Notch signaling pathways (*Figure 5—figure supplement 1C*), while KGA expression promoted mitophagy signaling pathway (*Figure 5—figure supplement 1D*). In SKNAS cells, GAC expression promoted PI3K-AKT and Hippo pathways (*Figure 5—figure supplement 1E*) and KGA expression promoted TNF signaling pathway (*Figure 5—figure supplement 1*). Thus, while GAC and KGA catalyzes the same chemical reaction by converting glutamine to glutamate, they also induce non-redundant cellular responses in neuroblastoma cells, suggesting that both isoforms might be important for neuroblastoma cells. Indeed, knockdown of both isoforms (GLS) showed a greater cancer cell killing phenotype than that of GAC or KGA alone did (*Figure 5G–I*).

## JMJD6 forms an interaction network with proteins involved in splicing and protein synthesis

To understand the mechanism of JMJD6 in regulating splicing in neuroblastoma, we performed an unbiased identification of JMJD6-interacting partners by introducing a FLAG-tagged JMJD6 into SK-N-AS and BE2C cells, followed by immunoprecipitation to pull down the JMJD6-associated complex and protein identification with mass spectrometry (*Figure 6A*). We found that JMJD6 mainly interacted with two classes of proteins which are involved in RNA splicing and protein synthesis in both cell lines, respectively (*Figure 6A*, *Figure 6—figure supplement 1*). We then validated the interactions of JMJD6 with splicing factors using immunoprecipitation and western blot, and demonstrated that JMJD6 formed a complex with these RNA binding proteins, including RBM39 (*Figure 6B*), a therapeutic target of high-risk neuroblastoma (*Singh et al., 2021*). Since JMJD6 also interacted with

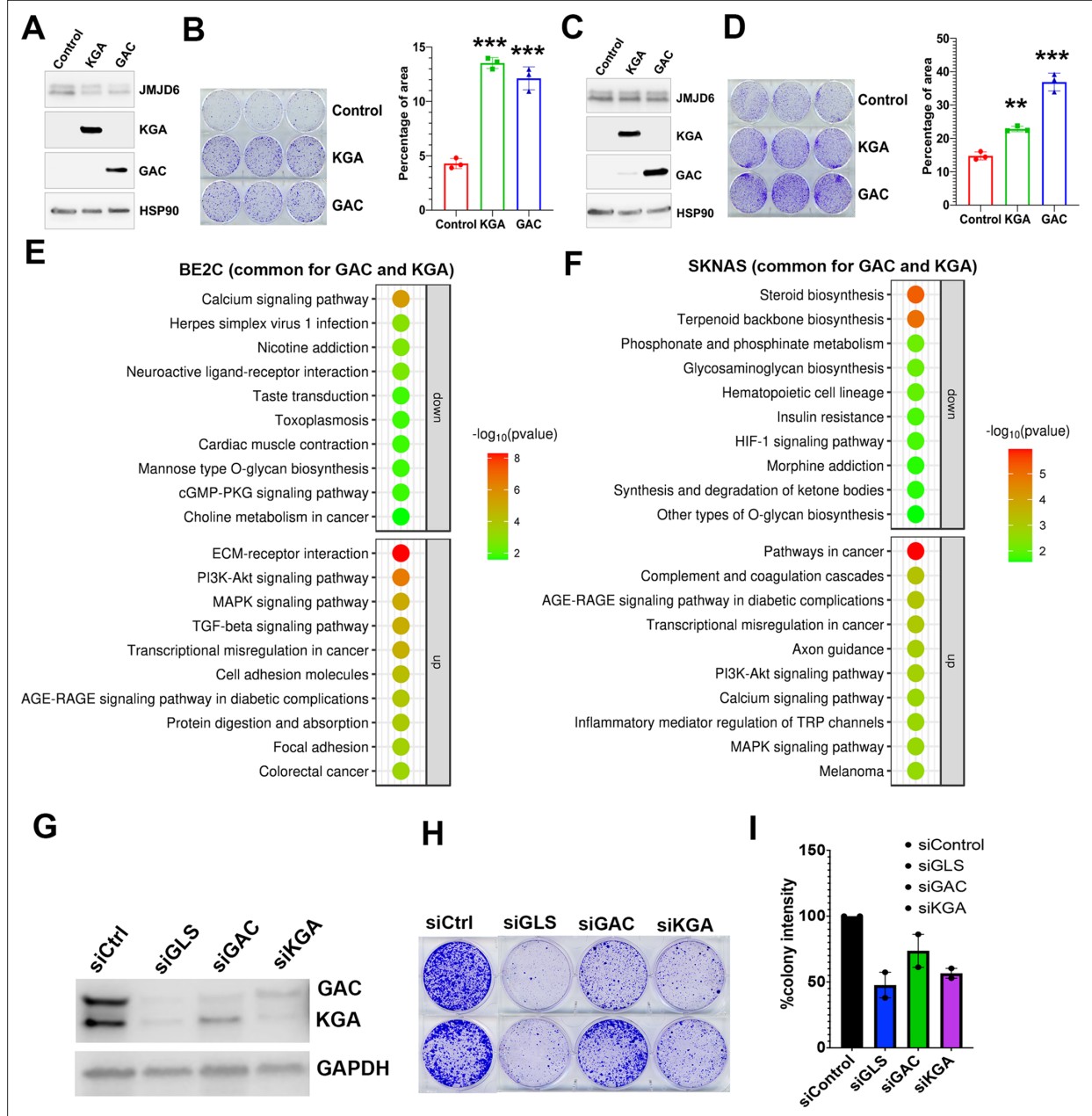

**Figure 5.** Glutaminase C (GAC) and kidney-type glutaminase (KGA) are both important for cell survival. (**A**) Western blotting analysis of expression of KGA and GAC in BE2C cells with indicated antibodies. (**B**) Colony formation assay of BE2C cells overexpressing KGA and GAC for 7 days (left = crystal violet staining, right = quantification of cell density). n=3 per group. ***p<0.001. (**C**) Western blotting analysis of expression of KGA and GAC in SK-N-AS cells with indicated antibodies. (**D**) Colony formation assay of SK-N-AS cells overexpressing KGA and GAC for 7 days (left = crystal violet staining, right = quantification of cell density). n=3 per group. **p<0.01, ***p<0.001. (**E**) Bubble blot showing the pathways significantly upregulated and downregulated by both KGA and GAC in BE2C cells. (**F**) Bubble blot showing the pathways significantly upregulated and downregulated by both KGA and GAC in SKNAS cells. (**G**) Whole cell lysates (on 72 hr) subject to western blot showing the knockdown of glutaminase (GLS) (both GAC and KGA), GAC alone, and KGA alone in BE2C cells. (**H**) Colony-forming assay (on day 7) of BE2C cells with knockdown of GLS (both GAC and KGA), GAC alone, and KGA alone. n=2 independent experiments. (**I**) Quantification of colonies in (H) using ImageJ software. n=2 independent experiments. Data are Mean ± SEM.

The online version of this article includes the following figure supplement(s) for figure 5:

**Figure supplement 1.** Upregulated and downregulated pathways by GAC and KGA.

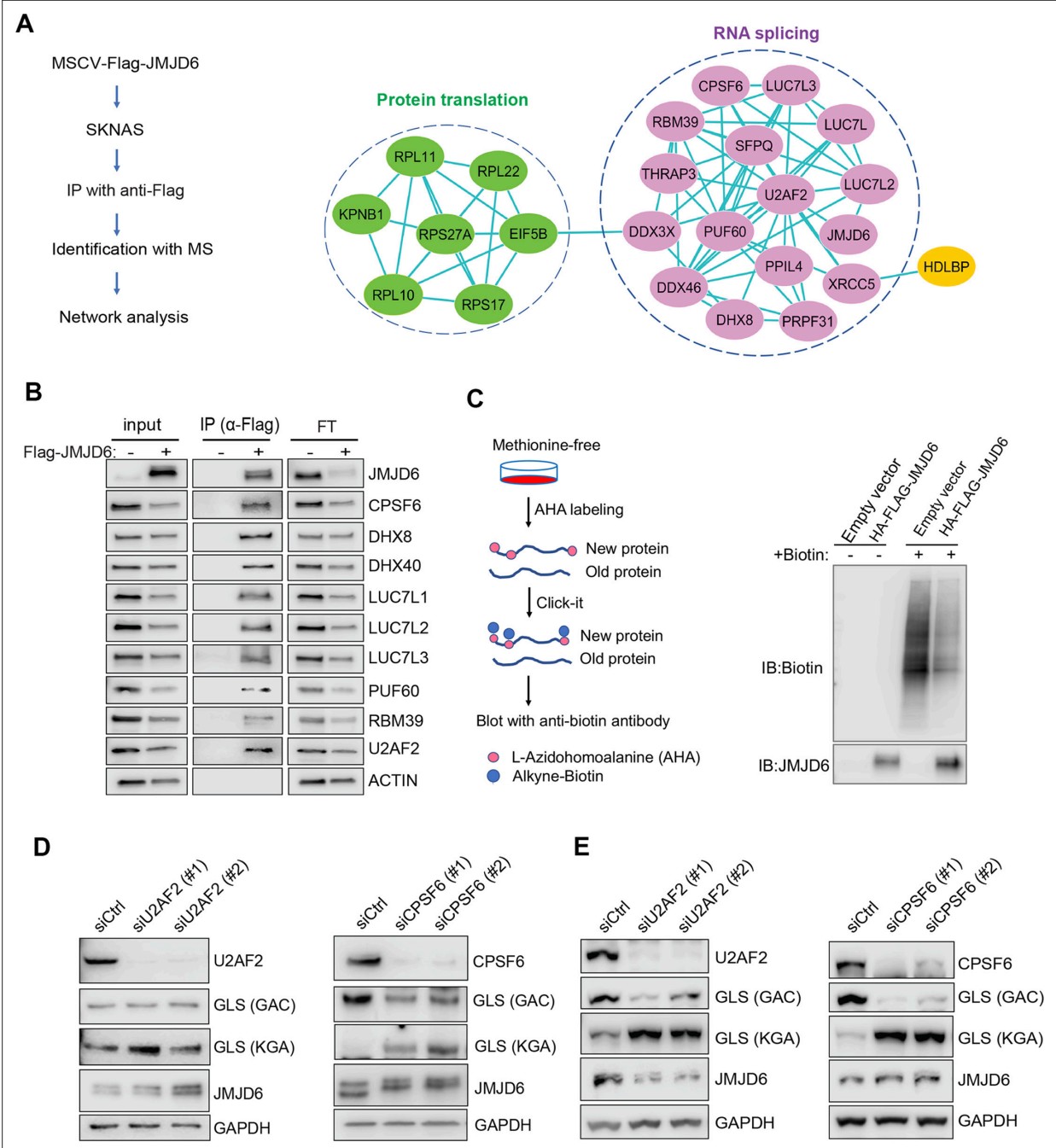

**Figure 6.** JMJD6 forms an interaction network that consists of proteins involved in splicing and protein synthesis. (**A**) FLAG-tagged JMJD6 transduced into SK-N-AS cells for immunoprecipitation with anti-FLAG followed by protein identification by mass spectrometry. The interacting protein partners of JMJD6 are analyzed by STRING protein network. (**B**) Immunoprecipitation followed by western blot to validate the JMJD6-interacting partners in SK-N-AS cells. IP = immunoprecipitation, FT = flowthrough. n=single experiment. (**C**) Click-iT AHA labeling showing the newly synthesized proteins after overexpression of JMJD6 in SK-N-AS cells. n=single experiment. (**D, E**) Western blot showing the expression of GAC and KGA isoforms in SKNAS (**D**), BE2C (**E**), after U2AF2 and CPSF6 knockdown for 72 hr. n=single experiment.

The online version of this article includes the following source data and figure supplement(s) for figure 6:

**Source data 1.** Mass spectrometric analysis JMJD6 interactomes in BE2C and SKNAS cells.

**Figure supplement 1.** The JMJD6 interactome in BE2C cells.

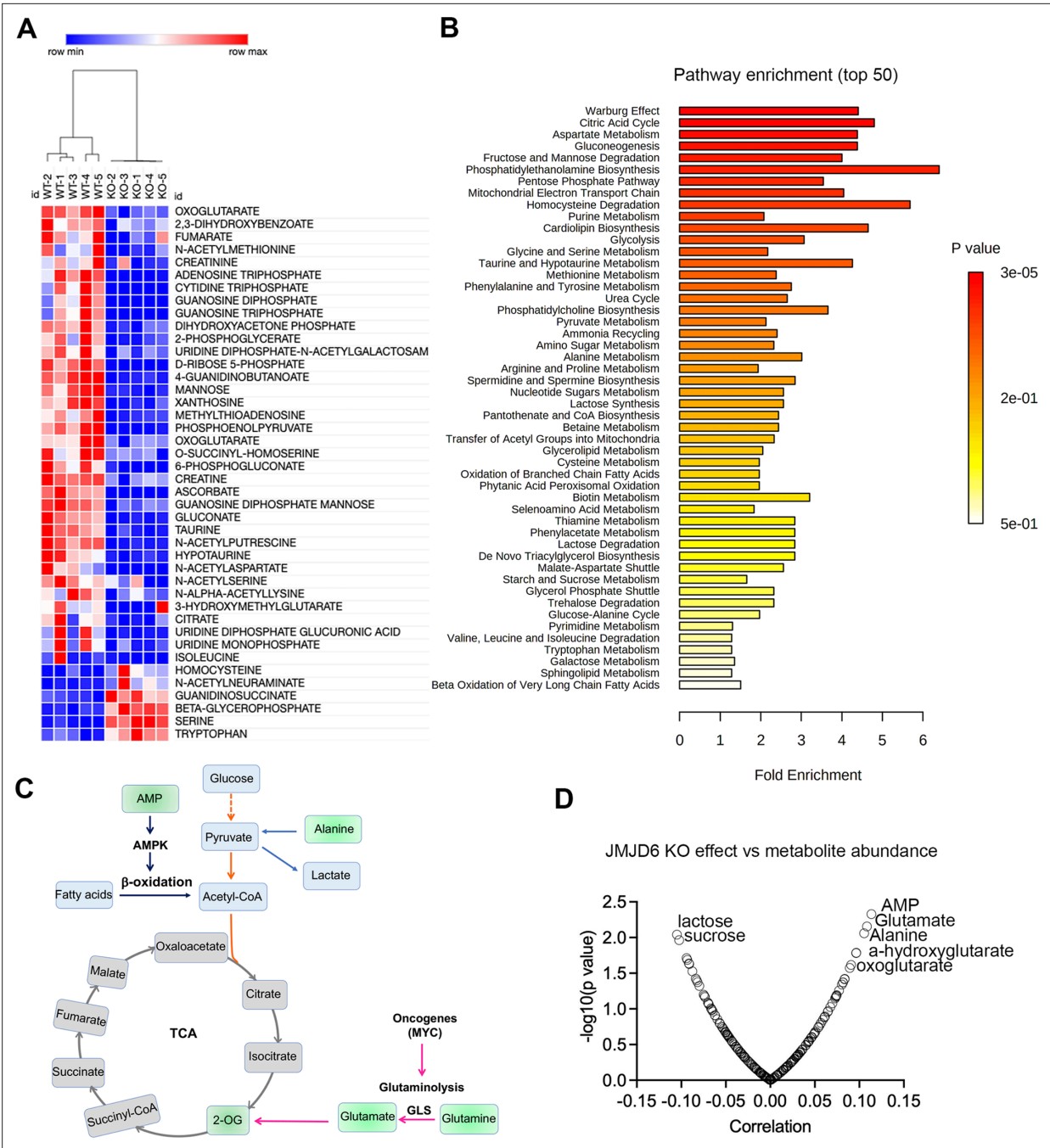

**Figure 7.** JMJD6 regulates production of citric acid cycle intermediates and NTP. (**A**) Heatmap showing the metabolites differentially expressed in SK-N-AS cells (n=5) after JMJD6 knockout (n=5) based on liquid chromatography with tandem mass spectrometry (LC-MS/MS) analysis. (**B**) Pathway analysis of metabolites downregulated by JMJD6 knockout. (**C**) Pathway cartoon showing the connections of tricarboxylic acid (TCA), glycolysis, glutaminolysis, and β-oxidation. (**D**) Correlation of metabolite abundance with JMJD6 dependency. The positive correlation indicates that the higher the abundance of metabolites, the more resistance of cells to JMJD6 knockout. On the contrary, the negative correlation indicates the higher the abundance of metabolites, the more sensitive of cells to JMJD6 knockout.

The online version of this article includes the following figure supplement(s) for figure 7:

**Figure supplement 1.** CRISPR knockout of JMJD6 in SKNAS cells.

several molecules involved in protein translation, we investigated if JMJD6 also regulates protein synthesis by using an approach, Click-iT AHA, to label the newly synthesized proteins, followed by western blot assessment (*Figure 6C*). Interestingly, overexpression of JMJD6 greatly reduced total protein synthesis (*Figure 6C*), suggesting that JMJD6 may antagonize protein production.

Then, we determined if the splicing factors interacting with JMJD6 also regulate GLS isoform expression. Among the splicing factors with which JMJD6 interacted, CPSF6 has been previously shown to regulate the alternative splicing of *GLS* (*Masamha et al., 2016*). We validated the function of CPSF6 in neuroblastoma cells and found that, indeed, loss of CPSF6 led to a dramatic switch from GAC to KGA in BE2C and SK-N-AS cells (*Figure 6D and E*). Previous studies showed that JMJD6 and U2AF2 (U2AF65) interact to regulate splicing (*Webby et al., 2009*; *Yi et al., 2017*). We also found that knockdown of U2AF2 resulted in a similar phenotype to JMJD6 knockdown in that the expression of KGA isoform was greatly increased in both cell lines (*Figure 6D and E*). These data collectively support the functions of JMJD6 in regulating the splicing of metabolic genes and protein homeostasis in MYC-driven neuroblastoma.

## JMJD6 regulates production of TCA intermediates and nucleoside triphosphate

To further dissect the biological consequences of loss of function of JMJD6, we created stable JMJD6 knockout clones (*Figure 7—figure supplement 1*) and defined the metabolite spectrum affected by loss of function by using liquid chromatography with tandem mass spectrometry (LC-MS/MS). JMJD6 knockout greatly reduced the production of TCA cycle intermediates (i.e. oxoglutarate, fumarate) and nucleoside triphosphate (ATP, CTP, GTP) (*Figure 7A*), indicating that JMJD6 is a key bioenergetic regulator in cancer cells. Pathway analysis revealed that the reduced metabolites were involved in the Warburg effect, TCA, pentose phosphate pathway, and mitochondrial electron transport chain (*Figure 7B*), all of which are critical for providing cancer cell bioenergetics for proliferation and survival. Oxoglutarate (α-ketoglutarate) and fumarate are downstream products of glutaminolysis (*Figure 7c*). We reasoned that cellular metabolites such as glutamate and oxoglutarate may predict the cytotoxic effect of loss of function of JMJD6. If cells have higher levels of glutamate and oxoglutarate, they might be less dependent on JMJD6 due to their higher capacity of buffering against reduced glutaminolysis. To test this hypothesis, we used DepMap data that included 225 metabolites in 928 cancer cell lines from over 20 lineages (*Li et al., 2019*), and analyzed the correlation of each metabolite with *JMJD6* gene dependency. The data showed that cells with high levels of AMP, glutamate, alanine, 2-hydroxyglutarate, and 2-oxoglutarate were more resistant to JMJD6 knockout, while cells with high levels of lactose and sucrose were more sensitive to JMJD6 knockout (*Figure 7C*). High levels of AMP activate AMP kinase, consequently leading to enhanced fatty acid oxidation to stimulate ATP production while alanine can be converted to pyruvate to provide acetyl-CoA to fuel the TCA cycle (*Figure 7C*). Therefore, high levels of AMP and alanine may provide cells alternative bioenergetics sources for survival. These data further indicate that JMDJ6 function is wired into the regulation of mitochondrial metabolism.

## JMJD6 determines the efficacy of indisulam, a molecular glue degrading splicing factor RBM39

Dysregulated splicing as a vulnerability of MYC-driven cancers provides a rationale to target neuroblastoma by using splicing inhibitors as a therapeutic approach. We and others have recently reported that indisulam, a 'molecular glue' that selectively degrades the splicing factor RBM39, is exceptionally effective at causing tumor regression in multiple high-risk neuroblastoma models without overt toxicity (*Singh et al., 2021*; *Nijhuis et al., 2022*), suggesting indisulam has translational potential. Understanding the factors determining the efficacy of indisulam or any other drug is critical for developing precision therapy, combination therapy, or preventing therapy resistance. In addition to complexing together (*Figure 6A and B*), *JMJD6* and *RBM39* exhibit significant correlation of co-dependency in cancer cells, namely, cancer cells have similar dependency on *JMJD6* and *RBM39* for survival (*Figure 8A*). These data indicate that JMJD6 may play a role in modulating the effect of indisulam. To test this hypothesis, we performed GSEA to identify the differential pathways between indisulam-sensitive and indisulam-less-sensitive neuroblastoma cells. It turned out that histone lysine demethylase (HDM) genes, including *JMJD6*, are present in the most significantly enriched gene signature in

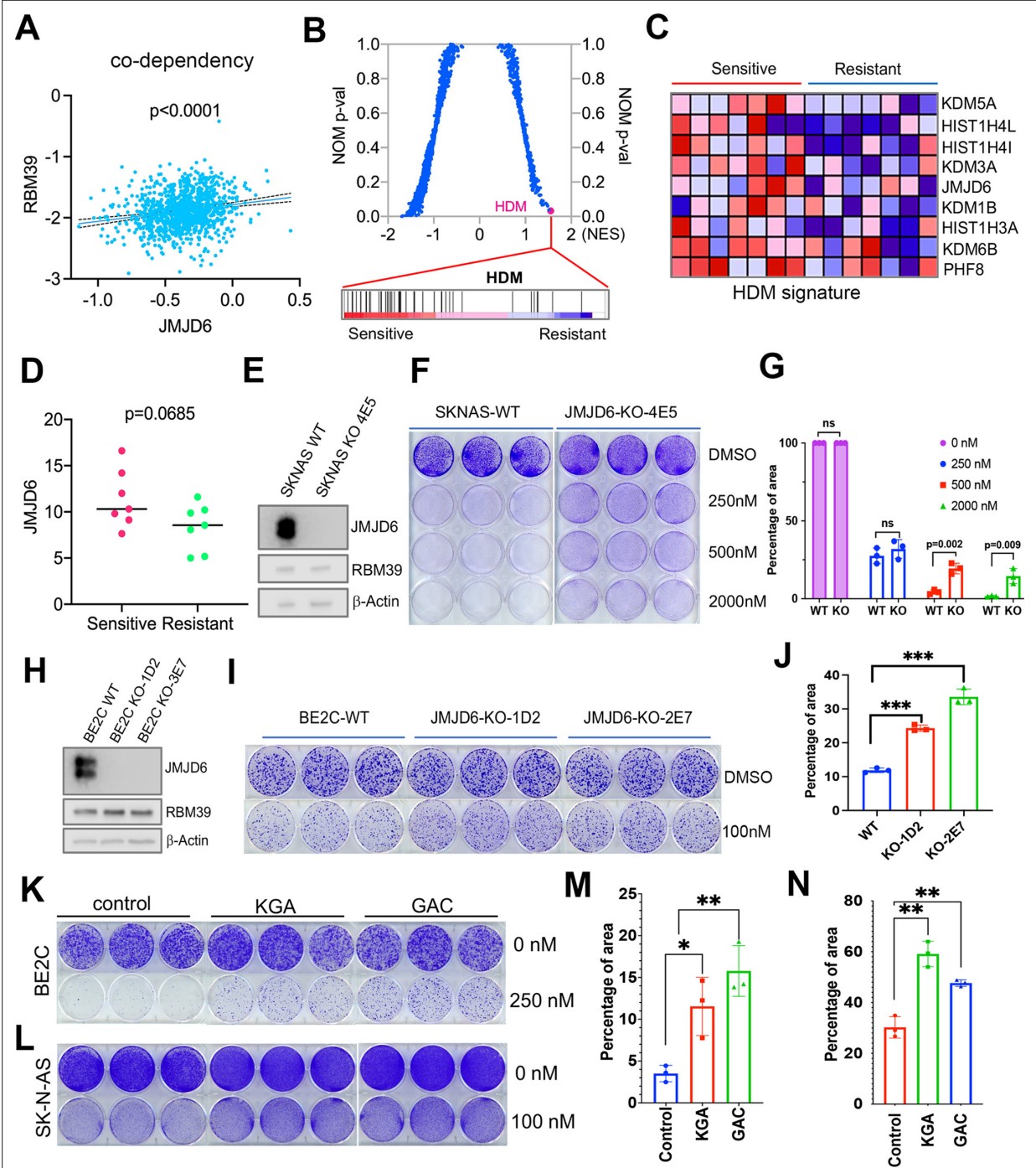

**Figure 8.** JMJD6-GAC pathway regulates the response of neuroblastoma cells to indisulam treatment. (**A**) Spearman correlation of effects of JMJD6 knockout and RBM39 knockout demonstrating the co-dependency of JMJD6 and RBM39 from DepMap CRISPR screening data (n=1086). Each dot represents one cell line. (**B**) Gene set enrichment analysis (GSEA) for indisulam sensitive vs resistant neuroblastoma cell lines based on CTD2 (Cancer Target Discovery and Development) data showing histone lysine demethylase gene signature is the one that is significantly associated with indisulam response. (**C**) Heatmap from GSEA (**B**) showing the individual genes in indisulam-sensitive and -resistant cells. (**D**) JMJD6 expression in indisulam-sensitive and -resistant neuroblastoma cells. p-Value calculated by Student's t-test. (**E**) Western blot showing JMJD6 knockout in SK-N-AS cells using indicated antibodies. (**F**) Colony formation of SK-N-AS cells in triplicates with or without JMJD6 knockout treated with different concentrations of indisulam for 7 days, stained with crystal violet. n=3 per group. (**G**) Quantification of cell density by using ImageJ software from (**F**) (n=triplicates). ns = not significant. **p<0.001, ***p<0.0001. (**H**) Western blot showing JMJD6 knockout in BE2C cells using indicated antibodies. (**I**) Colony formation of BE2C cells in triplicates with or without JMJD6 knockout treated with 100 nM of indisulam for 5 days, stained with crystal violet. n=3 per group. (**J**) Quantification of cell density by using ImageJ software from (I) (n=triplicates). ns = not significant. **p<0.001, ***p<0.0001. (**K**) Colony formation

*Figure 8 continued*

of BE2C cells in triplicates with KGA and GAC overexpression treated with 250 nM of indisulam for 5 days, stained with crystal violet. n=3 per group. (**L**) Colony formation of SK-N-AS cells in triplicates with KGA and GAC overexpression treated with 100 nM of indisulam for 7 days, stained with crystal violet. (**M**) Quantification of cell density by using ImageJ software from (K) (n=triplicates). *p<0.01, **p<0.001. (**N**) Quantification of cell density by using ImageJ software from (L) (n=triplicates). **p<0.001. Data are Mean ± SEM.

indisulam-sensitive cells (*Figure 8B, C, and D*). Indeed, knockout of *JMJD6* led to partial but significant resistance to indisulam treatment of SK-N-AS cells (*Figure 8E–G*) and BE2C cells (*Figure 8H–J*), supporting that cells with high JMJD6 expression are more dependent on RBM39. Since we found that JMJD6 plays a key role in modulating glutaminolysis, we tested if expression of GAC or KGA could affect the activity of indisulam. Interestingly, overexpression of either GAC and KGA renders BE2C and SK-N-AS cells more resistant to indisulam treatment (*Figure 8K–N*), suggesting that enhanced glutaminolysis may confer therapeutic resistance to spliceosome inhibition.

## Discussion

MYC is an oncogenic driver of many types of cancer and plays a pivotal role in regulating glycolysis, glutaminolysis, nucleotide and lipid synthesis, and ribosome and mitochondrial biogenesis (*Stine et al., 2015*). Recent studies have revealed that there is also an interplay between MYC and pre-mRNA splicing machinery (*Hsu et al., 2015*; *Hirsch et al., 2015*; *Koh et al., 2015*; *Phillips et al., 2020*; *David et al., 2010*; *Rauch et al., 2011*; *Ge et al., 2019*; *Seton-Rogers, 2015*). Pre-mRNA splicing is an essential biological process catalyzed by the spliceosome to produce mature mRNAs (*Matera and Wang, 2014*; *Wahl et al., 2009*). Over 90% of multiexon genes in the human genome undergo alternative splicing (*Pan et al., 2008*; *Wang et al., 2008*), which significantly expands the diversity of the proteome and consequently impacts various biological functions (*Yang et al., 2016*; *Liu et al., 2017*). In this study, we found that the chromosome 17q locus, which is frequently gained in MYC-driven cancers, houses numerous genes essential to cancer cell survival that are implicated in pre-mRNA splicing, ribosome and mitochondrial biogenesis, and other biological functions. Particularly, JMJD6, which is located on 17q25 and is amplified in a number of cancer types, physically interacts with a subset of splicing factors such as RBM39 and regulates the alternative splicing of metabolic genes. Depletion of JMJD6 inhibits cancer cell proliferation and impedes tumor growth while overexpression of JMJD6 promotes MYC-mediated tumorigenesis, suggesting that JMJD6 and potentially other 17q genes have oncogenic functions in cellular transformation. Previous studies suggest that JMJD6 and BRD4 interact to regulate gene transcription (*Wong et al., 2019*). However, our unbiased identification of the JMJD6 interactome only identified a subset of proteins involved in mRNA splicing and protein translation in neuroblastoma cells, suggesting that JMD6 may predominantly regulate protein homeostasis to facilitate MYC-mediated transformation. Nevertheless, we cannot exclude the possibility that our immunoprecipitation conditions were too stringent, leading to dissociation of proteins that loosely or dynamically bind to JMJD6.

MYC-induced metabolic reprogramming triggers cellular dependency on exogenous glutamine as a source of carbon for mitochondrial membrane potential maintenance and macromolecular synthesis (*Pavlova and Thompson, 2016*), leading to 'glutamine addiction' (*Pavlova and Thompson, 2016*). Glutaminolysis is a process by which GLS and GLS2 convert glutamine to glutamate, which is, in turn, converted by glutamate dehydrogenase or transaminase to 2-oxoglutarate that is further catabolized in the TCA cycle. Additionally, glutamate is a substrate for production of glutathione, an important antioxidant. We previously showed that neuroblastoma relies on MYCN-induced glutaminolysis for survival (*Wang et al., 2018*). In this study, our RNA-seq barely detected the expression of *GLS2* in the neuroblastoma cell models we used, indicating that GLS is the major enzyme that catalyzes glutamine in these model systems. *GLS* has two isoforms, *GAC* and *KGA*, resulting from alternative splicing. KGA is mainly localized in the cytoplasm while GAC is localized in mitochondria and has a higher basal activity (*Cassago et al., 2012*). *GAC* mRNA levels strongly correlate with the conversion of glutamine to glutamate, as a proxy for GAC activity (*Daemen et al., 2018*). The positive correlation of *JMJD6* and *GAC* suggests that the JMJD6-high tumors have enhanced glutaminolysis activity. CSPF6 and the noncoding RNA *CCAT2* have been reported to regulate the splicing of GLS isoforms (*Masamha et al., 2016*; *Redis et al., 2016*). Interestingly, we found that depletion of JMJD6 leads to a *GLS*

isoform switch from *GAC* to *KGA*, indicating that JMJD6 is involved in alternative splicing of *GLS*. We further found that JMJD6 physically interacts with CPSF6 in a splicing network and validated that loss of CPSF6 results in remarkable induction of the KGA isoform. We further validated that other splicing factors such as U2AF2 also interact with JMJD6 to regulate the GLS isoform switch. These data indicate that cancer cells can adjust metabolism through alternative splicing to produce enzymes with distinct subcellular localization and activity that promote cellular transformation or progression of an oncogenic phenotype. The cooperation of JMJD6 and MYC in cellular transformation further supports the hypothesis that JMJD6 is needed for metabolic reprogramming triggered by MYC. However, overexpression of either GAC or KGA promotes cell proliferation, suggesting that the switching of KGA/GAC is a cellular fitness mechanism in response to interruption of the spliceosome by adjusting the metabolic rate. Within the tumor microenvironment (i.e. replete and deplete oxygen and nutrient supply) GLS activity is possibly finely tuned through splicing mechanism for adaption.

Additionally, we found that JMJD6 physically interacts with a subset of ribosomal proteins that are responsible for protein translation. Interestingly, overexpression of JMJD6 reduces global protein synthesis. A recent study showed that MYC overactivation leads to proteotoxic stress in cells by enhancing global protein synthesis, consequently causing cell death (*Gong et al., 2021*). The increased global protein synthesis by MYC needs to be buffered through loss of DDX3X, a regulator

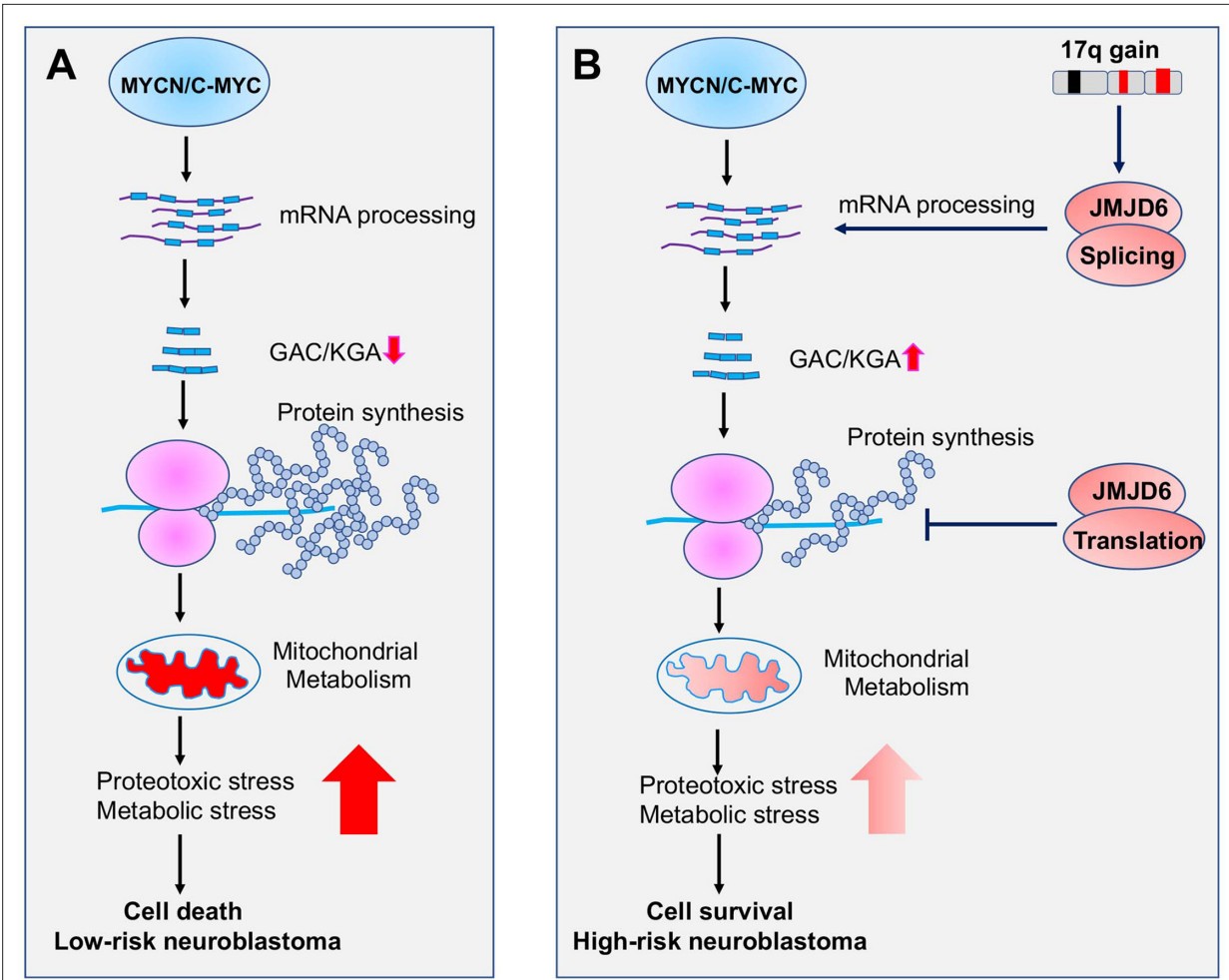

**Figure 9.** Working mechanism of JMJD6 in MYC-driven neuroblastoma. Overactive MYC drives high-load of gene transcription, enhanced protein synthesis, and high rate of metabolism, leading to detrimental cellular stresses and consequent cell death (Model **A**). However, when 17q is amplified, high levels of JMJD6 and other proteins encoded by 17q genes physically interact with the splicing and translational machineries, enhancing pre-mRNA splicing of metabolic genes such as glutaminase (GLS) and inhibiting global protein synthesis, respectively, leading to reduced detrimental stresses and enhanced cancer cell survival and tumorigenesis (Model **B**). The high levels of JMJD6 predicts high dependency of RBM39, which are more sensitive to indisulam treatment.

of ribosome biogenesis and global protein synthesis, for lymphomagenesis (*Gong et al., 2021*). Our findings suggest that, besides the functions in regulating alternative splicing for metabolism, JMJD6 is involved in MYC-mediated cell transformation by buffering unwanted proteotoxic stress due to high rate of protein synthesis induced by MYC (*Figure 9*).

Neuroblastoma is responsible for as much as 15% of childhood cancer mortality (*Bosse and Maris, 2016*). With current intensive multimodal therapies, 5-year survival rates for high-risk patients remain less than 50% (*Pinto et al., 2015*; *Brodeur, 2003*; *Maris et al., 2007*; *Cohn et al., 2009*). In addition, survivors of high-risk disease have a significant risk of developing long-term side effects including subsequent malignant neoplasms due to cytotoxic chemotherapy and radiotherapy (*Suh et al., 2020*; *Nathan et al., 2007*). Unfortunately, developing effective precision therapies against high-risk neuroblastoma has been challenging due to the lack of targetable recurrent mutations in neuroblastoma (*Pugh et al., 2013*; *Molenaar et al., 2012*; *Brady et al., 2020*). Our previous study showed that indisulam, the splicing inhibitor that targets RBM39, a JMJD6 interacting partner, induced a durable complete response in multiple high-risk neuroblastoma models, supporting its potential use in future clinical trials. Our current study showed that JMJD6 expression is positively correlated with the effect of indisulam, and knockout of JMJD6 confers resistance to indisulam treatment. In line with the biological functions of JMDJ6 in regulating GLS isoform expression and mitochondrial metabolism, overexpression of GAC or KGA also caused resistance to indisulam treatment. These data indicate that JMJD6 could serve as a biomarker that predicts response to indisulam or other splicing inhibitors.

## Limitation of the study

Our study focused on the understanding of JMJD6 function in neuroblastoma cell lines. In the future, we will consolidate our study by expanding our models to patient-derived xenografts, organoids, and neuroblastoma genetic models, in comparison with non-cancerous cells. Although we have identified a conserved interactome of JMJD6 in neuroblastoma cells, it remains to be determined whether it is neuroblastoma-specific and essential to MYC-driven cancers. The genome-wide RNA binding by JMJD6 in cancer cells and normal cells coupled with isotope labeling to dissect the metabolic effect of JMJD6 will enhance our understanding of the biological functions of JMJD6, awaiting future studies. Inability to target the enhanced pre-mRNA splicing of metabolic genes in MYC-driven cancer cells by pharmacological inhibition of JMJD6 is another limitation, due to lack of selective and potent JMJD6 inhibitors.

## Materials and methods
### Cell lines

KELLY (ECACC, 92110411), SIMA (DSMZ, ACC164), BE2C (ATCC, CRL2268), IMR32 (ATCC, CCL127), SK-N-AS (ATCC, CRL2137), CHLA20 (COG) were cultured in 1× RPMI1640 (Corning, 15-040-CV) supplemented with 10% fetal bovine serum (Sigma-Aldrich, F2442), 1% L-glutamine (Corning, A2916801). NIH3T3 (ATCC, CRL1658) and 293T (ATCC, CRL3216) cells were cultured in 1 DMEM supplemented with 10% fetal bovine serum (Sigma-Aldrich, F2442), 1% L-glutamine (Corning, A2916801). All cells were maintained at 37°C in an atmosphere of 5% $CO_2$. JoMa1 cells kindly provided by Dr Schulte (Department of Pediatric Oncology and Hematology, University Children's Hospital Essen, Essen, Germany) were cultured in NCC Medium: DMEM (4.5 mg/ml glucose, L-glutamine, pyruvate): Ham's F12 (1:1) was supplemented with: 1% N2-Supplement (Invitrogen, no. 17502-048), 2% B27-Supplement (Invitrogen, no. 17504-044), 10 ng/mL EGF (Invitrogen), 1 ng/mL FGF (Invitrogen), 100 U/mL Penicillin-Streptomycin (Invitrogen), and 10% Chick-Embryo-Extract (Gemini Bio-Products, CA, USA). Neural crest culture medium was supplemented with 200 nM 4-OH-tamoxifen (Sigma no. H7904) in routine culture to ensure nuclear localization of c-MycERT and JoMa1 cell proliferation. JoMa1 cells were grown on cell culture flask/dish coated with fibronectin, NCC Medium supplemented with 200 nm 4-OHT was changed daily. Cells were passaged after 3–4 days in culture when 70% confluence was reached ($4 \times 10^6$ cells/10 cm dish).

All human-derived cell lines were validated by short tandem repeat profiling using PowerPlex 16 HS System (Promega) once a month. Additionally, a PCR-based method was used to screen for mycoplasma once a month employing the LookOut Mycoplasma PCR Detection Kit (MP0035,

Sigma-Aldrich) and JumpStart Taq DNA Polymerase (D9307, Sigma-Aldrich) to ensure cells were free of mycoplasma contamination.

## Antibodies

GAPDH (Cell Signaling Technology, 5174s, rabbit antibody), MYCN (Santa Cruz Biotechnology, 53993, mouse antibody), FLAG (Sigma, F1804, mouse antibody), Biotin (Bethyl Laboratories, A150-109A, rabbit antibody), ACTIN (Sigma, A3854, mouse antibody), PUF60 (Thermo Fisher, PA5-21411, rabbit antibody), U2AF2 (Novus Biologicals, NBP2-04140, rabbit antibody), CPSF6 (Bethyl Laboratories, 357A, rabbit antibody), DHX40 (Novus Biologicals, NBP1-91834, rabbit antibody), DHX8 (Abcam, AB181074, rabbit antibody), LUC7L1 (Novus Biologicals, NBP2-56401, rabbit antibody), LUC7L2 (Novus Biologicals, NBP2-33621, rabbit antibody), LUC7L3 (Novus Biologicals, NBP1-88053, rabbit antibody), RBM39 (ATLAS, HPA001519, rabbit antibody), GLS (KGA-specific) (Proteintech, 20170-1-AP, rabbit antibody), GLS (GAC-specific) (Proteintech, 19958-1-AP, rabbit antibody), JMJD6 (ATLAS, HPA059156, rabbit antibody), JMJD6 (Santa Cruz Biotechnology, sc-28348, mouse antibody).

## Retroviral plasmids and retrovirus packaging

MSCV-IRES-GFP and MSCV-IRES-mCherry were obtained from St Jude Vector Core. Human JMJD6 and murine MYCN were subcloned into MSCV-IRES-GFP and MSCV-IRES-mCherry, respectively. The MSCV-CMV-CMV-FLAG-HA-JMJD6 was purchased from Addgene (Addgene # plasmid 31358). The retrovirus packaging was done as described in the following procedure. Briefly, HEK93T cells were transfected with viral vectors by combining 5 µg of target vector, 4.4 µg of pMD-old-gag-pol, and 0.6 µg of VSV-G plasmids in 400 µL of DMEM without serum or L-glutamine. PEIpro transfection reagent (Polyplus 115-010) was added at 2:1 (PEIpro µL:µg of plasmid) per 100 mm dish of cells and mixed well, and incubated at room temperature for at least 20 min, prior to adding cells. The following day, fresh medium was added to cells. For 3–4 days, viral media was harvested and replaced twice per day. Viral media was centrifuged at 1500 RPM for 10 min and filtered through a 0.45 µm vacuum filter. Virus was concentrated by ultracentrifugation at 28.5 kRPM for 2 hr at 4°C, aspirated, and resuspended in either OptiMEM or phosphate-buffered saline (PBS), aliquoted, and frozen at –80°C until use. cDNAs for GAC and KGA were synthesized by GenScript and cloned into pGenLenti vector for virus packaging.

## siRNA transfection

25 µM of each siRNA oligo was resuspended in 500 µL of prewarmed Opti-MEM, reduced serum medium (Gibco Life Technologies # 31985-070) in six-well plates. To each well, 7 µL of RNAiMax (Invitrogen Lipofectamine RNAiMAX transfection reagent 13778100) was added, mixed, and left at room temperature for 10 min. After incubation, 100,000 cells of each indicated cell line were added to each well in a total of 2 mL volume with RPMI medium supplemented with 10% FBS. JMJD6 siRNA#63, 5-CCAAAGUUAUCAAGGAAA-3; JMJD6 siRNA#75, 5-CAGUGAAGAUGAAGAUGAA-3. U2AF2 siRNA#1 AGAAGAAGAAGGUCCGU; U2AF2 siRNA#2 GUGGCAGUUUCAUAUUUG. CPSF6 siRNA#1 GGAUCACCUUCCAAGACA. CPSF6 siRNA#2 AGAACCGUCAUGACGAUU. GLS1 siRNA, CAACTGGCCAAATTCAGTC; GAC siRNA, CCTCTGTTCTGTCAGAGTT; KGA siRNA, ACAGCGGG ACTATGATTCT.

## SDS-PAGE and western blot

Cells were washed twice with ice-cold PBS and directly lysed on ice with 2× sample loading buffer (0.1 M Tris HCl [pH 6.8], 200 mM dithiothreitol [DTT], 0.01% bromophenol blue, 4% sodium dodecyl sulfate [SDS], and 20% glycerol). On ice, cell lysates were sonicated once with a 5 s bursts at 40% amplitude output (Sonics, VIBRA CELL) followed by 25 min heating at 95°C. After the cell lysates were centrifuged at 13,000×g at room temperature for 2 min, 10–20 µL of the cell lysates were separated on 4–15% Mini-PROTEAN TGX Stain-Free Protein Gels from Bio-Rad and transferred to methanol-soaked polyvinylidene difluoride membranes (Millipore). Lysates for RBM39 G268V mutant cell lines and DCAF15 genetically modified cells were generated as previously described (*Han et al., 2017*). Membranes were blocked in PBS buffer supplemented with 0.1% Tween 20 and 5% skim milk (PBS-T) and incubated for 1 hr at room temperature under gentle horizontal shaking. Membranes were incubated overnight at 4°C with the primary antibodies. The next day, membranes were washed three

times (for 5 min) with PBS-T at room temperature. Protected from light, membranes were then incubated with goat anti-mouse or goat anti-rabbit HRP-conjugated secondary antibodies (1:5000) for 1 hr at room temperature, followed by three 5 min washes with PBS-T at room temperature. Lastly, membranes were incubated for 1 min at room temperature with SuperSignal West Pico PLUS Chemiluminescent Substrate (34580, Thermo Fisher Scientific) and the bound antigen-antibody complexes were visualized using Odyssey Fc Imaging System (LI-COR Corp., Lincoln, NE, USA).

## RNA extraction and RT-PCR isoforms of GLS

RNA was extracted using RNeasy Plus Mini Kit (QIAGEN, reference # 74136) following the manufacturer's protocol. cDNA was prepared in 20 µL reaction from 500 ng of total RNA using Superscript IV First Strand Synthesis System (Invitrogen, reference # 1809105) kit. RT-PCRs were run in triplicates (n=3) in the 7500 Real-Time PCR system by Applied Biosystems (Thermo Fisher Scientific) using power SYBR Green PCR master mix (Applied Biosystems, reference # 4367660). ΔΔCT methods were applied to analyze the results. The following primers were used to perform the quantitative Real-Time PCR- GAPDH (Forward: AACGGGAAGCTTGTCATCAATGGAAA, Reverse: GCATCAGCAGAGGGGG CAGAG), GAC (Forward: GAGGTGCTGGCCAAAAAGCCT, Reverse: AGGCATTCGGTTGCCCAAAC T), KGA (Forward: CTGCAGAGGGTCATGTTGAA, Reverse: ATCCATGGGAGTGTTATTCCA).

## Lentiviral packaging of pLenti and shRNA

The GAC and KGA cDNAs were synthesized by Genscript company and cloned into pGenLenti vector. The TRC lentiviral-based shRNA knockdown plasmids for JMJD6 were purchased from Horizon Discovery (sh#46: RHS3979-201781036, TTAAACCAGGTAATAGCTTCG; sh#47: RHS3979-201781037, ATCTTCACTGAGTAGCCATCG). The lentiviral shJMJD6 and shControl (pLKO.1) particles were packaged by Vector Lab at St Jude. Briefly, HEK293T cells were transfected with shRNA constructs and helper plasmids (pCAG-kGP1-1R, pCAG4-RTR2, and pHDM-G). The 48 and 72 hr post-transfection replication-incompetent lentiviral particles were harvested and transduced into cells with 8 µg/mL of polybrene. 48 hr later, 1 µg/mL of puromycin was added for selection for additional 48 hr before injection into mice or immunoblotting.

## JMJD6 CRISPR knockout method

Genetically modified neuroblastoma cells were generated by using CRISPR-Cas9 technology. Briefly, 400,000 NB cells were transiently co-transfected with 100 pmol of chemically modified gRNA (GGAC TCTGGAGCGCCTAAAA) (Synthego), 33 pmol of Cas9 protein (St Jude Protein Production Core), 200 ng of pMaxGFP (Lonza), and, using solution P3 and program DS-150 in small cuvettes according to the manufacturer's recommended protocol. Five days post-nucleofection, cells were sorted for GFP+ (transfected) cells and plated as single cells into 96-well plates. Cells were clonally expanded and screened for the desired modification using targeted next-generation sequencing followed by analysis with CRIS.py (https://pubmed.ncbi.nlm.nih.gov/30862905/).

## Colony formation of JoMa1 cells

Matrigel was kept at 4°C to being liquified for 6 hr. 50 µL of Matrigel per 1 cm$^2$ area was added to 24-well plate without air bubbles. The 24-well plate was kept at 37°C in cell culture incubator till it was solidified. 200 of JoMa1 cells transduced with GFP, JMJD6, MYCN, MYCN+JMJD6 in DMEM:F12 enriched media without tamoxifen were seeded onto the 24-well coated with Matrigel. This was done in triplicate. Cells were checked daily, and media were changed every 3 days without disturbing the Matrigel by removing and adding media gently. To stain the colonies, cells were fixed by formaldehyde (3.7% in PBS) for 2 min at room temperature, followed by permeabilization with 100% methanol (not ice-cold) for 20 min at room temperature. The colonies were stained by 0.4% crystal violet.

## Crystal violet staining

After removing media, cells were washed with Dulbecco's PBS (DPBS) without calcium or magnesium (DPBS, Lonza) and treated with 4% formaldehyde in PBS (paraformaldehyde [PFA]) for 20 min. Once PFA was removed, cells were stained with 0.1% crystal violet stain for 1 hr. KGA/GAC overexpression colony formation: 5000 cells were plated in BE2C control, KGA and GAC overexpressing cells and were cultured for 7 days; 10,000 cells were plated in SKNAS control, KGA and GAC

overexpressing cells and were cultured for 7 days (n=3). After 7 days, medium was removed and cells were washed with DPBS (Lonza) and treated with 4% formaldehyde in PBS [PFA] for 30 min. PFA was later removed and cells were stained with 0.1% crystal violet stain for 1 hr. Experiments were repeated twice. Indisulam treatment on WT and JMJD6 knockout cell lines: JMJD6-WT and knockout cells were plated in 12-well plate (50,000 cells/well for SKNAS) and 6-well plate (5000 cells/well for BE2C) (n=3). Next day, cells were treated with indisulam with indicated concentration for 7 days (SKNAS cells) and 5 days (BE2C cells). Crystal violet staining was performed to visualize and quantify the colony formation. Experiments were repeated twice. Indisulam treatment on KGA/GAC overexpressing cell lines: 10,000 BE2C and 100,000 SKNAS control, KGA and GAC overexpressing cells were plated in six-well plate (n=3). Next day, cells were treated with indisulam for 7 days. Crystal violet staining was performed to visualize and quantify the colony formation. Experiments were repeated twice.

## Click-iT AHA labeling assay for metabolic labeling of newly synthesized proteins

Click-iT was performed as previously described (*Hu et al., 2018*). Briefly, cells were plated at 5 million cells per 100 mm dish in RPMI supplemented with 10% FBS. Cells were washed with warm PBS and replaced with methionine-free medium (Thermo 21013024 supplement with glutamine and sodium pyruvate) for 1 hr at 37°C in 5% $CO_2$. Following, fresh methionine-free media containing 50 µM of Click-iT AHA (L-azidohomoalanine) (Thermo C10102) was added to the cells for 2 hr at 37°C. After AHA labeling, cells were washed with warm PBS and lysed with 1% SDS, 50 mM Tris-HCl, (pH 8.0) supplemented with phosphatase inhibitors (PhosSTOP, Sigma) and protease inhibitors (cOmplete Mini, Roche) by applying the buffer directly to the plate, incubating the cells on ice for 30 min, tilting the plates, and collecting the lysate. Lysates were briefly sonicated, vortexed for 5 min, and centrifuged at 18,000×*g* for 5 min at 4°C. Total protein quantification was assayed using the EZQ Protein Quantification Kit (Thermo R33200) according to the manufacturer's protocol and results were read on a fluorescence-based microplate reader (BioTek Synergy 2). Click chemistry of the biotin-alkyne (PEG4 carboxamide-propargyl biotin) (Thermo B10185) to the AHA-labeled lysates was performed using the Click-iT Protein Reaction Buffer Kit (Thermo C10276) using a concentration of 40 µM biotin-alkyne per click reaction (and no biotin-alkyne added for controls). Following the click reaction, samples were either assayed for total biotinylated protein by following the manufacturer's protocol. For total biotinylated protein, briefly, 600 µL of methanol, 150 µL of chloroform, and 400 µL of megaOhm water was sequentially added and vortexed, followed by centrifugation at 18,000×*g* for 5 min. The upper aqueous phase was discarded, and 450 µL of methanol was added, vortexed, and centrifuged again at 18,000×*g* for 5 min. This methanol step was performed in duplicate to remove residual reaction components. Protein pellets were allowed to air-dry and resuspended in a suitable volume of sample buffer and heated prior to western blot analysis.

## Immunoprecipitation

$5×10^6$ BE2C and SK-N-AS cells expressing FLAG-JMJD6 were cultured in 150 cm dish with complete RPMI media. Cells were washed twice with cold PBS after reaching 95% confluency, then lysed in 1 mL lysis buffer 50 mM Tris-HCl, pH 7.4, 150 mM NaCl, 1 mM EDTA, 1% Triton X-100 with complete protease inhibitors (Sigma 11836170001, added fresh) and PhosSTOP (Sigma 4906845001). Cells were scrapped into a 1.5 mL Eppendorf tube and incubated on ice for 15 min, which were mixed by vortex every 5 min. Cell lysates were spun by 13,500 RPM for 10 min at 4°C. The supernatant was transferred to a new tube. The cell lysates were subject to immunoprecipitation using M2 anti-FLAG beads (Sigma, M8823) overnight by rocking at 4°C. The following day, beads were washed 3× with buffer and eluted with 5 packed gel volumes of FLAG peptide in TBS buffer (3 µL of stock FLAG peptide at 5 µg/µL per 100 µL of TBS buffer) while rotating at 4°C for 30 min. Beads were briefly spun and the supernatant was removed from the beads (eluate). This elution step was repeated one more time and pooled with the first eluate. Prior to western blot, input, flowthroughs, and elution samples were processed by adding 4× sample buffer supplemented with 50 mM DTT and heated at 75°C for 10 min prior to running on a gel.

## RNA immunoprecipitation

SK-N-AS cells expressing FLAG-JMJD6 were grown in a 10 cm dish in RPMI complete media. After 70% confluency, cells were washed with cold PBS twice and then were subject to lysis with Polysome Lysis Buffer (100 mM KCl, 5 mM MgCl$_2$, 10 mM HEPES, pH 7.0, 0.5% NP-40, 1 mM DTT, 100 U/mL RNasin RNase inhibitor [Promega, N2511], 2 mM vanadyl ribonucleoside complexes solution [Sigma, 94742], 25 μL protease inhibitor cocktail for mammalian cells [Sigma, P8340]). Cell lysates were precleared with magnetic IgG beads for 1 hr. The cell lysates were subject to immunoprecipitation using M2 anti-FLAG beads (Sigma, M8823) overnight by rocking at 4°C. The same amounts of lysates were saved at –80°C for input RNA extraction. The beads were washed with 250 μL Polysome Lysis Buffer for four times, flowed by washing with Polysome Lysis Buffer containing 1M urea. RNA was released by adding 150 μL of Polysome Lysis Buffer containing 0.1% SDS and 45 μg proteinase K (Ambion, AM2548) and incubated at 50°C for 30 min. RNA was extracted with phenol-chloroform-isoamyl alcohol mixture (Sigma, 77618). RNA was recovered by adding 2 μL of GlycoBlue (15 mg/mL, Ambion, AM9516), 36 μL of 3 M sodium acetate and 750 μL ethanol followed by incubation at –20°C for overnight. RNA was precipitated with 70% ethanol and air-dried, followed by resuspension with RNase-free water followed by DNaseI (Promega, M6101) treatment to remove genomic DNA. The resultant RNAs were subjected to RT-qPCR analysis using three sets of GAC and KGA primers and 18S rRNA as control. 18S rRNA F: GCTTAATTTGACTCAACACGGGA; 18S rRNA R: AGCTATCAATCTGTCAATCCTGTC. GLS-GACiso_F: GAGGTGCTGGCCAAAAAGCCT; GLS-GACiso_R: AGGCATTCGGTTGCCCAAACT. GLS-KGAiso_F: CTGCAGAGGGTCATGTTGAA; GLS-KGAiso_R: ATCCATGGGAGTGTTATTCCA. KGA_set2_F: GCAGCCTCCAGGTGCTTTCA; KGA_set2_R: GTAATGGGAGGGCAGTGGCA. KGA_set3_F: TGCCCGACACTGCCCTTTAG; KGA_set3_R: CCTGCCAGACAGACAACAGCA. GAC_set2_F: TGCTTCTCAAGGCCTTACTGC; GAC_set2_R: AGGCATTCGGTTGCCCAAACT. GAC_set3_F: CCTTCTAGAGGTGCTGGCCAAA; GAC_set3_R: TGCAACACAAATATGCAGTAAGGC. For validation of protein immunoprecipitation, 20% of beads after overnight incubation were removed and processed as follows: Beads were washed 3× with buffer and eluted with 5 packed gel volumes of FLAG peptide in TBS buffer (3 μL of stock FLAG peptide at 5 μg/μL per 100 μL of TBS buffer) while rotating at 4°C for 30 min. Beads were briefly spun and the supernatant was removed from the beads (eluate). This elution step was repeated one more time and pooled with the first eluate.

## Identification of JMJD6-interacting partners by LC-MS/MS

Protein samples were run on a short gel as described in a previously published protocol (*Xu et al., 2009*). Proteins in the gel bands were reduced with DTT (Sigma) and alkylated by iodoacetamide (Sigma). The gel bands were then washed, dried, and rehydrated with a buffer containing trypsin (Promega). Samples were digested overnight, acidified, and the resulting peptides were extracted. The extracts were dried and reconstituted in 5% formic acid. The peptide samples were loaded on a nanoscale capillary reverse phase C18 column by an HPLC system (Thermo EASY-nLC 1000) and eluted by a gradient. The eluted peptides were ionized and detected by a mass spectrometer (Thermo LTQ Orbitrap Elite). The MS and MS/MS spectra were collected over a 90 min liquid chromatography gradient. Database searches were performed using Sequest (v28, revision 13) search engine against a composite target/decoy Uniprot human protein database. All matched MS/MS spectra were filtered by mass accuracy and matching scores to reduce protein false discovery rate to <1%. Spectral counts matching to individual proteins reflect their relative abundance in one sample after the protein size is normalized. The spectral counts between samples for a given protein were used to calculate the p-value based on G-test (*Bai et al., 2013*).

## Metabolome profiling by LC-MS/MS

JMJD6 knockout or parental SK-N-AS cells were cultured in six-well plates to ~85% confluence and washed with 2 mL ice-cold 1× PBS. The cells were then harvested in 300 μL freezing 80% acetonitrile (vol/vol) into 1.5 mL tubes and lysed in the presence of 0.5 mm zirconia/silica beads by Bullet Blender (Next Advance) at 4°C until the samples were homogenized. The resulting lysate was then centrifuged at 21,000×*g* for 5 min and the supernatant was dried by SpeedVac. The samples were resuspended in 50 μL of 1% acetonitrile plus 0.1% trifluoroacetic acid, and separated by Ultra-C18 Micro spin columns (Harvard apparatus) into hydrophilic metabolites (flowthrough) and hydrophobic metabolites (eluent of 125 μL of 80% acetonitrile plus 0.1% trifluoroacetic acid). Ten μL of hydrophilic

metabolites were dried, reconstituted in 3 µL of 66% acetonitrile, and analyzed by a ZIC-HILIC column (150×2.1 mm$^2$, EMD Millipore) coupled with a Q Exactive HF Orbitrap MS (Thermo Fisher) in negative mode and metabolites were eluted within a 45 min gradient (buffer A: 10 mM ammonium acetate in 90% acetonitrile [pH = 8]; buffer B: 10 mM ammonium acetate in 100% H$_2$O [pH = 8]). Twenty µL of hydrophobic metabolites were dried and resuspended in 3 µL of 5% formic acid followed by separation with a self-packed nanoC18 column (75 µm×15 cm with 1.9 µm C18 resin from Dr Maisch GmbH) and detected with a Q Exactive HF Orbitrap MS (Thermo Fisher) in positive mode. Metabolites were eluted within a 50 min gradient (buffer A: 0.2% formic acid in H$_2$O; buffer B: 0.2% formic acid in acetonitrile). MS settings for both types of samples included MS1 scans (120,000 resolution, 100–1000 mass/charge [m/z], 3×10$^6$ AGC, and 50 ms maximal ion time) and 20 data-dependent MS2 scans (30,000 resolution, 2×10$^5$ AGC, ~45 ms maximal ion time, HCD, Stepped NCE [50, 100, 150], and 20 s dynamic exclusion). A mix of all samples served as quality control was injected in the beginning, middle, and the end of the samples to monitor the signal stability of the instrument. The data analysis was performed by a recently developed software suite JUMPm. Raw files were converted to mzXML format followed by peak feature detection for individual sample and feature alignment across samples. Metabolite identification was supported by matching the retention time, accurate m/z ratio, and MS/MS fragmentation data to our in-house authentic compound library and the matching of m/z and MS/MS fragmentation data, to downloaded experimental MS/MS library (MoNA, https://mona.fiehnlab.ucdavis.edu/), in silico database generated from Human Metabolome Database (HMDB), and mzCloud (https://mzcloud.org). Peak intensities were used for metabolite quantification. The data was normalized by both cell numbers (before data collection) and trimmed median intensity of all features across samples (post data collection).

## Differential gene expression and GSEA for RNA-seq experiments

Total RNA from cells and tumor tissues were performed using the RNeasy Mini Kit (QIAGEN) according to the manufacturer's instructions. Paired-end sequencing was performed using the High-Seq platform with 100 bp read length. Total stranded RNA-seq data were processed by the internal Auto-Mapper pipeline. Briefly, the raw reads were firs trimmed (Trim-Galore v0.60), mapped to human genome assembly (GRCh38) (STAR v2.7), and then the gene-level values were quantified (RSEM v1.31) based on GENCODE annotation (v31). Low count genes were removed from analysis using a CPM cutoff corresponding to a count of 10 reads and only confidently annotated (level 1 and 2 gene annotation) and protein-coding genes are used for differential expression analysis. Normalization factors were generated using the TMM method, counts were then transformed using voom, and transformed counts were analyzed using the lmFit and eBayes functions (R limma package v3.42.2). The significantly up- and downregulated genes were defined by at least twofold changes and adjusted p-value < 0.05. Then GSEA was carried out using gene-level log2 fold changes from differential expression results against gene sets in the Molecular Signatures Database (MSigDB 6.2) (gsea2 v2.2.3).

## RNA splicing analysis

After mapping RNA-seq data, rMATS v4.1.0 was used for RNA alternative splicing analysis by using the mapped BAM files as input. Specifically, five different kinds of alternative splicing events were identified, i.e., skipped exon, alternative 5'-splicing site, alternative 3'-splicing site, mutually exclusive exon, and intron retention. To keep consistent, the same GTF annotation reference file for mapping was used for rMATS. For stranded RNA-seq data, the argument '--libType fr-firststrand' was applied. To process reads with variable lengths, the argument '--variable-read-length' was also used for rMATS. To select statistically significantly differential splicing events, the following thresholds were used: FDR < 0.05 and the absolute value of IncLevelDifference > 0.1. For visualization, the IGV Genome Browser was used to show the sashimi plots of splicing events. To investigate the genome-wide correlations of differential splicing between two genotypes (e.g. shRNA knockdown of JMJD6 and non-target shRNA in cells), we extracted splice junctions for all samples of both genotypes of interest from the STAR (*Dobin et al., 2013*) output files suffixed with 'SJ.out.tab', which contain high confidence collapsed splice junctions. Only those unique mapped reads crossing the junctions were considered. By extracting the union of the unique junction positions, we constructed a unified junction-read feature vector for each sample. Then, we normalized the junction-read vectors of each sample with TMM method in 'voom' and 'limma' and R package, assuming a negative binomial distribution. Next,

we averaged the junction-read vectors for samples of the same genotype. The gene-level expression was estimated based on the canonical junctions from the most abundant isoforms estimated for each gene. The fold changes of exon junctions significantly deviated from gene-level changes were regarded as differentially spliced junctions for between cell-line comparisons.

### Data mining

*JMJD6, GAC, and KGA* expression in tumor tissues were downloaded from R2 (https://portals.broadinstitute.org/ccle), Kocak dataset GSE45547 (649 samples), and Fisher dataset GSE120572 (394 samples). In both datasets, the probe UKv4_A_23_P311616 represented *JMJD6*, the probe UKv4_A_23_P308800 represented *GAC,* and UKv4_A_23_P39766 represented *KGA. JMJD6* expression data from the RNA-seq data of various pediatric cancer tissues were downloaded from St Jude cloud (https://pecan.stjude.cloud/). The copy number alterations of *JMJD6* and the related Kaplan-Meier analysis were downloaded from cBioportal (https://www.cbioportal.org/). The data for correlation of metabolite abundance and JMJD6 knockout effect were downloaded from DepMap (https://depmap.org/portal/).

### Pathway network analysis

The 114 essential fitness genes to neuroblastoma cell survival identified through genome-wide CRISPR/Cas9 library screen were uploaded into STRING program (https://string-db.org) for network interaction analysis with confidence threshold 0.15. The resulting network was then uploaded into Cytoscape program for presentation (*Shannon et al., 2003*). The clusters were grouped based on the biological functions of each gene.

### Copy number analysis of JMJD6 and other genes encoding JmjC domain histone demethylases from St Jude neuroblastoma cohort

Somatic copy number alternations were determined by CONSERTING (PMID: 25938371) for each pair of tumor and normal samples. The normalize read depth ratio (log2 ratio) for the CNV segments with JmjC domain containing proteins were extracted and used for CNV heatmap generation (https://CRAN.R-project.org/package=pheatmap; *Kolde, 2019*) and hierarchical clustering of samples.

### Xenograft studies

All murine experiments were done in accordance with a protocol (JY: 615) approved by the Institutional Animal Care and Use Committee of St Jude Children's Research Hospital. (1) shRNA-mediated JMJD6 knockdown. Neuroblastoma cells were transduced with shRNA lentiviral particles targeting JMJD6. 48 hr later, 1 µg/mL of puromycin was added for selection for additional 48 hr. Cancer cells ($5 \times 10^6$) were mixed with Matrigel (1:1 ratio in volume) and subcutaneously injected into the flank sites of NSG mice. (2) JMJD6 and MYC-mediated transformation. After JoMa1 cells were transduced with GFP, JMJD6, MYCN, and JMJD6/MYCN, $10^4$ cells per group were mixed with Matrigel (1:1 ratio in volume) and subcutaneously injected into the flank sites of 4–6 weeks of female NSG mice (JAX, RRID:IMSR_JAX:005557). Mice were sacrificed when they reached the humane endpoint. Tumors were measured by using electronic calipers, and volumes calculated as width π/6×d3, where d is the mean of two diameters taken at right angles.

### Statistical analysis

All quantitative data are presented as mean ± SD. Unpaired Student's t-test was performed for comparison of two groups. Spearman correlation was used to assess the relationship between two variables. Kaplan-Meier method was used to estimate the survival rate. Mann-Whitney rank test (two-sided) was used to compare the tumor volume between two groups at every time point. p-Values across multiple time points were adjusted for multiple comparison using the Holm-Sidak method. $p<0.05$ was considered as statistically significant. All the statistical analyses, except where otherwise noted, were performed using GraphPad Prism (v9).

### Materials availability

The plasmids expressing JMJD6, MYCN, GAC, and KGA generated in this study will be freely distributed to non-profit in accordance with the guidelines about the sharing of unique research resources.

Requests from for-profit corporations will be negotiated by the Office of Technology Management of St Jude Children's Research Hospital.

## Acknowledgements

We thank the staff of the St Jude Animal Resource Center and Hartwell Center for their dedication and expertise. We thank COG for providing cell lines. The work was supported by American Cancer Society-Research Scholar (130421-RSG-17-071-01-TBG, JY), National Cancer Institute (1R01CA229739-01, JY, R01CA266600, JY). The work was also supported by the American Lebanese Syrian Associated Charities (ALSAC). The content is solely the responsibility of the authors and does not necessarily represent the official views of the National Institutes of Health.

## Additional information

### Funding

| Funder | Grant reference number | Author |
|---|---|---|
| American Cancer Society | 130421-RSG-17-071-01-TBG | Jun Yang |
| National Cancer Institute | 1R01CA229739-01 | Jun Yang |
| National Cancer Institute | R01CA266600 | Jun Yang |
| American Lebanese Syrian Associated Charities | | Jun Yang |

The funders had no role in study design, data collection and interpretation, or the decision to submit the work for publication.

### Author contributions

Carolyn M Jablonowski, Waise Quarni, Data curation, Formal analysis, Validation, Investigation, Visualization, Methodology; Shivendra Singh, Jie Fang, Vishwajeeth Pagala, Investigation; Haiyan Tan, Dhanushka Hewa Bostanthirige, Hongjian Jin, David Finkelstein, Gang Wu, Data curation, Formal analysis, Investigation, Methodology; Ti-Cheng Chang, Ji-Hoon Cho, Formal analysis; Dongli Hu, Formal analysis, Investigation; Sadie Miki Sakurada, Resources; Shondra M Pruett-Miller, Resources, Formal analysis, Methodology; Ruoning Wang, Kevin Freeman, Resources, Writing – review and editing; Andrew Murphy, Writing – review and editing; Junmin Peng, Supervision; Andrew M Davidoff, Supervision, Writing – review and editing; Jun Yang, Data curation, Formal analysis, Supervision, Visualization, Writing – review and editing, Conceptualization, Funding acquisition, Writing – original draft, Project administration

### Author ORCIDs

Shondra M Pruett-Miller ⓘ http://orcid.org/0000-0002-3793-585X
Ruoning Wang ⓘ http://orcid.org/0000-0001-9798-8032
Andrew Murphy ⓘ http://orcid.org/0000-0001-6747-0355
Jun Yang ⓘ https://orcid.org/0000-0002-4233-3220

### Ethics

This study was performed in strict accordance with the recommendations in the Guide for the Care and Use of Laboratory Animals of the National Institutes of Health. All of the animals were handled according to approved institutional animal care and use committee (IACUC) protocol (#615) of St Jude Children's Research Hospital.

Reviewer #1 (Public Review): https://doi.org/10.7554/eLife.90993.3.sa1
Reviewer #2 (Public Review): https://doi.org/10.7554/eLife.90993.3.sa2
Author Response https://doi.org/10.7554/eLife.90993.3.sa3

## Additional files

### Supplementary files
- MDAR checklist
- Source data 1. JMJD6 interactome in BE2C cells and related pathways.

### Data availability

All data generated during this study are included in the manuscript. Uncropped immunoblots and images are accessible as source data. RNA-seq data are deposited in NCBI GEO under GSE185867 and GSE248283.

The following datasets were generated:

| Author(s) | Year | Dataset title | Dataset URL | Database and Identifier |
|---|---|---|---|---|
| Yang J, Finkelstein D | 2023 | Metabolic reprogramming of cancer cells by JMJD6-mediated alternative splicing | https://www.ncbi.nlm.nih.gov/geo/query/acc.cgi?acc=GSE185867 | NCBI Gene Expression Omnibus, GSE185867 |
| Quarni W, Fang J, Jin H, Yang J | 2024 | Metabolic reprogramming of cancer cells by JMJD6-mediated pre-mRNA splicing is associated with therapeutic response to splicing inhibitor | https://www.ncbi.nlm.nih.gov/geo/query/acc.cgi?&acc=GSE248283 | NCBI Gene Expression Omnibus, GSE248283 |

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

# Appendix 1

**Appendix 1—key resources table**

| Reagent type (species) or resource | Designation | Source or reference | Identifiers | Additional information |
|---|---|---|---|---|
| Antibody | Anti-GAPDH (rabbit polyclonal) | Cell Signaling Technology | | WB 1:1000 |
| Antibody | Anti-MYCN (mouse monoclonal) | Santa Cruz Biotechnology | 53993, RRID:AB_831602 | WB 1:1000 |
| Antibody | Anti-FLAG (mouse monoclonal) | Sigma | F1804, RRID:AB_262044 | WB 1:1000 |
| Antibody | Anti-Biotin (rabbit polyclonal) | Bethyl Laboratories | A150-109A, RRID:AB_67327 | WB 1:1000 |
| Antibody | Anti-ACTIN (mouse monoclonal) | Sigma | A3854, RRID:AB_262011 | WB 1:1000 |
| Antibody | Anti-PUF60 (rabbit polyclonal) | Thermo Fisher | PA5-21411, RRID:AB_11154782 | WB 1:1000 |
| Antibody | Anti-U2AF2 (rabbit polyclonal) | Novus Biologicals | NBP2-04140 | WB 1:1000 |
| Antibody | Anti-CPSF6 (rabbit polyclonal) | Bethyl Laboratories | A301-357A, RRID:AB_937785 | WB 1:1000 |
| Antibody | Anti-DHX40 (rabbit polyclonal) | Novus Biologicals | NBP1-91834, RRID:AB_11040145 | WB 1:1000 |
| Antibody | Anti-DHX8 (rabbit recombinant monoclonal) | Abcam | AB181074 | WB 1:1000 |
| Antibody | Anti-LUC7L1 (rabbit polyclonal) | Novus Biologicals | NBP2-56401 | WB 1:1000 |
| Antibody | Anti-LUC7L2 (rabbit polyclonal) | Novus Biologicals | NBP2-33621 | WB 1:1000 |
| Antibody | Anti-LUC7L3 (rabbit polyclonal) | Novus Biologicals | NBP1-88033, RRID:AB_11030257 | WB 1:1000 |
| Antibody | Anti-RBM39 (rabbit polyclonal) | ATLAS | HPA0015191, RRID:AB_1079749 | WB 1:1000 |
| Antibody | Anti-GLS (KGA-specific) (rabbit polyclonal) | Proteintech | 20170–1-AP, RRID:AB_10665373 | WB 1:1000 |
| Antibody | Anti-GLS (GAC-specific) (rabbit polyclonal) | Proteintech | 19958–1-AP, RRID:AB_10640899 | WB 1:1000 |
| Antibody | Anti-JMJD6 (rabbit polyclonal) | ATLAS | HPA059156, RRID:AB_2683934 | WB 1:1000 |
| Antibody | Anti-JMJD6 (mouse monoclonal) | Santa Cruz Biotechnology | sc-28348, RRID:AB_628185 | WB 1:1000 |
| Antibody | M2 anti-FLAG beads (mouse monoclonal) | Sigma | M8823, RRID:AB_2637089 | Antibody-conjugated beads |
| Cell line (*Homo sapiens*) | KELLY | ECACC | 92110411, RRID:CVCL_2092 | Neuroblastoma cell line, human, pediatric |
| Cell line (*Homo sapiens*) | SIMA | DSMZ | ACC164, RRID:CVCL_1695 | Neuroblastoma cell line, human, pediatric |
| Cell line (*Homo sapiens*) | BE2C | ATCC | CRL2268, RRID:CVCL_0529 | Neuroblastoma cell line, human, pediatric |
| Cell line (*Homo sapiens*) | IMR32 | ATCC | CCL127, RRID:CVCL_0346 | Neuroblastoma cell line, human, pediatric |

*Appendix 1 Continued on next page*

| Reagent type (species) or resource | Designation | Source or reference | Identifiers | Additional information |
|---|---|---|---|---|
| Cell line (*Homo sapiens*) | SK-N-AS | ATCC | CRL2137, RRID:CVCL_6602 | Neuroblastoma cell line, human, pediatric |
| Cell line (*Homo sapiens*) | CHLA20 | COG | RRID:CVCL_6602 | Neuroblastoma cell line, human, pediatric |
| Cell line (*Homo sapiens*) | HEK293T/293T | ATCC | CRL3216, RRID:CVCL_0063 | Embryonic kidney, human |
| Cell line (*Mus musculus*) | NIH3T3 | ATCC | CRL1658, RRID:CVCL_0594 | Fibroblast cell line, mouse |
| Cell line (*Mus musculus*) | JoMa1 | Dr Schulte (Department of Pediatric Oncology and Hematology, University Children's Hospital Essen, Essen, Germany) | | Neural crest cell line, mouse |
| Chemical compound, drug | Indisulam | MedKoo Biosciences | MedKoo Cat#: 201540 | RBM39 inhibitor |
| Commercial assay or kit | PowerPlex 16 HS System | Promega | DC2101 | Used for short tandem repeat (STR) profiling of all human-derived cell lines |
| Commercial assay or kit | LookOut Mycoplasma PCR Detection Kit | Sigma-Aldrich | MP0035 | Used for mycoplasma screening for all cell lines |
| Commercial assay or kit | JumpStart Taq DNA Polymerase | Sigma-Aldrich | D9307 | Used for mycoplasma screening for all cell lines |
| Commercial assay or kit | Rneasy Plus Mini Kit | QIAGEN | 74136 | For isolating RNA from cells |
| Commercial assay or kit | Superscript IV First Strand Synthesis System | Invitrogen | 1809105 | Generating cDNA from RNA |
| Commercial assay or kit | PowerUp SYBR Green Master Mix | Applied Biosystems | A25743 | Master mix for real-time PCR |
| Commercial assay or kit | Click-iT AHA (L-azidohomoalanine) | Thermo Fisher | C10102 | Kit component used for Click-iT Metabolic labeling of nascent proteins |
| Commercial assay or kit | EZQ Protein Quantification Kit | Thermo Fisher | R33200 | Kit component used for Click-iT Metabolic labeling of nascent proteins |
| Commercial assay or kit | Biotin-alkyne (PEG4 carboxamide-propargyl biotin) | Thermo Fisher | B10185 | Kit component used for Click-iT Metabolic labeling of nascent proteins |
| Commercial assay or kit | PEIpro | Polyplus | 115--010 | Transfection reagent; used at 2:1 (μL:μg of DNA) |
| Commercial assay or kit | RNAiMAX | Invitrogen | 13778100 | RNAi transfection reagent; 7 μL used per 25 μM siRNA oligo |
| Gene (*Homo sapiens*) | JMJD6 | NCBI | NM_001081461.2 | |
| Gene (*Homo sapiens*) | GLS (GAC isoform) | NCBI | NM_014905.5 | |
| Gene (*Homo sapiens*) | GLS (KGA isoform) | NCBI | NM_001256310.2 | |
| Gene (*Mus musculus*) | Mycn | NCBI | NM_001293228.2 | |
| Other | N2-Supplement | Invitrogen | 17502-048 | Supplement neural crest culture medium for JoMa1 cells |
| Other | B27-Supplement | Invitrogen | 17504-044 | Supplement neural crest culture medium for JoMa1 cells |

| Reagent type (species) or resource | Designation | Source or reference | Identifiers | Additional information |
|---|---|---|---|---|
| Other | Chick-Embryo-Extract | Gemini Bio-Products | | Supplement neural crest culture medium for JoMa1 cells |
| Other | 4-OH-tamoxifen | Sigma | H7904 | Supplement neural crest culture medium to ensure nuclear localization of c-MycERT in JoMa1 cells |
| Other | cOmplete Protease Inhibitor | Sigma | 11836170001 | Protease inhibitor for immunoprecipitation |
| Other | PhosSTOP | Sigma | 4906845001 | Phosphatase inhibitor for immunoprecipitation |
| Other | FLAG peptide | St Jude | | Used for elution of FLAG-tagged peptides during immunoprecipitation; 3 µL of stock FLAG peptide at 5 µg/µL per 100 µL of TBS buffer |
| Other | Rnasin Rnase inhibitor | Promega | N2511 | Use at 100 U/mL for RNA immunoprecipitation |
| Other | Vanadyl ribonucleoside complexes solution | Sigma | 94742 | Use at 2 mM for RNA immunoprecipitation |
| Other | Proteinase K | Ambion | AM2548 | Digestion of protein in RNA immunoprecipitation |
| Other | Phenol-chloroform-isoamyl alcohol mixture | Sigma | 77618 | Precipitation of nucleotides in RNA immunoprecipitation |
| Other | GlycoBlue | Ambion | AM9516 | Recovery of RNA in RNA immunoprecipitation |
| Recombinant DNA reagent | MSCV-IRES-GFP (plasmid) | St Jude Vector Core | | |
| Recombinant DNA reagent | MSCV-IRES-mCherry (plasmid) | St Jude Vector Core | | |
| Recombinant DNA reagent | MSCV-JMJD6-IRES-GFP (plasmid) | This paper | | |
| Recombinant DNA reagent | MSCV-Mycn-IRES-mCherry (plasmid) | This paper | | |
| Recombinant DNA reagent | MSCV-CMV-CMV-FLAG-HA-JMJD6 (plasmid) | Addgene | 31358 | |
| Recombinant DNA reagent | pMD-old-gag-pol (plasmid) | St Jude Vector Core | | |
| Recombinant DNA reagent | VSV-G (plasmid) | St Jude Vector Core | | |
| Recombinant DNA reagent | pGenLenti (plasmid) | Genscript | | Lentiviral expression vector used to express cDNA sequence of either GAC or KGA isoform of GLS gene |
| Recombinant DNA reagent | TRC lentiviral-based shRNA knockdown plasmids to JMJD6 'sh#46' | Horizon Discovery | RHS3979-201781036 | TTAAACCAGGTAATAGCTTCG |
| Recombinant DNA reagent | TRC lentiviral-based shRNA knockdown plasmids to JMJD6 'sh#47' | Horizon Discovery | RHS3979-201781037 | ATCTTCACTGAGTAGCCATCG |
| Recombinant DNA reagent | shControl (pLKO.1) | St Jude Vector Core | | |
| Recombinant DNA reagent | Lentiviral helper plasmids | St Jude Vector Core | | pCAG-kGP1-1R |
| Recombinant DNA reagent | Lentiviral helper plasmids | St Jude Vector Core | | pCAG4-RTR2 |
| Recombinant DNA reagent | pMaxGFP | Lonza | | For CRISPR-Cas9-mediated editing |

| Reagent type (species) or resource | Designation | Source or reference | Identifiers | Additional information |
|---|---|---|---|---|
| Peptide, recombinant protein | EGF | Invitrogen | Recombinant protein, media supplement | EGF |
| Peptide, recombinant protein | FGF | Invitrogen | Recombinant protein, media supplement | FGF |
| Peptide, recombinant protein | Cas9 protein | St Jude Protein Production Core | Recombinant protein, peptide | Cas9 protein |
| Sequence-based reagent | 18S rRNA F | IDT | RT-PCR primers | GCTTAATTTGACTCAACACGGGA |
| Sequence-based reagent | 18S rRNA R | IDT | RT-PCR primers | AGCTATCAATCTGTCAATCCTGTC |
| Sequence-based reagent | GLS-GACiso_F | IDT | RT-PCR primers | GAGGTGCTGGCCAAAAAGCCT |
| Sequence-based reagent | GLS-GACiso_R | IDT | RT-PCR primers | AGGCATTCGGTTGCCCAAACT |
| Sequence-based reagent | GLS-KGAiso_F | IDT | RT-PCR primers | CTGCAGAGGGTCATGTTGAA |
| Sequence-based reagent | GLS-KGAiso_R | IDT | RT-PCR primers | ATCCATGGGAGTGTTATTCCA |
| Sequence-based reagent | KGA_set2_F | IDT | RT-PCR primers | GCAGCCTCCAGGTGCTTTCA |
| Sequence-based reagent | KGA_set2_R | IDT | RT-PCR primers | GTAATGGGAGGGCAGTGGCA |
| Sequence-based reagent | KGA_set3_F | IDT | RT-PCR primers | TGCCCGACACTGCCCTTTAG |
| Sequence-based reagent | KGA_set3_R | IDT | RT-PCR primers | CCTGCCAGACAGACAACAGCA |
| Sequence-based reagent | GAC_set2_F | IDT | RT-PCR primers | TGCTTCTCAAGGCCTTACTGC |
| Sequence-based reagent | GAC_set2_R | IDT | RT-PCR primers | AGGCATTCGGTTGCCCAAACT |
| Sequence-based reagent | GAC_set3_F | IDT | RT-PCR primers | CCTTCTAGAGGTGCTGGCCAAA |
| Sequence-based reagent | GAC_set3_R | IDT | RT-PCR primers | TGCAACACAAATATGCAGTAAGGC |
| Sequence-based reagent | siRNA to JMJD6 (#63) | Dharmacon | | CCAAAGUUAUCAAGGAAA |
| Sequence-based reagent | siRNA to JMJD6 (#75) | Dharmacon | | CAGUGAAGAUGAAGAUGAA |
| Sequence-based reagent | siRNA to U2AF2 (#1) | Dharmacon | | AGAAGAAGAAGGUCCGU |
| Sequence-based reagent | siRNA to U2AF2 (#2) | Dharmacon | | GUGGCAGUUUCAUAUUUG |
| Sequence-based reagent | siRNA to CPSF6 (#1) | Dharmacon | | GGAUCACCUUCCAAGACA |
| Sequence-based reagent | siRNA to CPSF6 (#2) | Dharmacon | | AGAACCGUCAUGACGAUU |
| Sequence-based reagent | siRNA to GLS | Dharmacon | | CAACTGGCCAAATTCAGTC |

| Reagent type (species) or resource | Designation | Source or reference | Identifiers | Additional information |
|---|---|---|---|---|
| Sequence-based reagent | siRNA to GLS (GAC-specific isoform) | Dharmacon | | CCTCTGTTCTGTCAGAGTT |
| Sequence-based reagent | siRNA to GLS (KGA-specific isoform) | Dharmacon | | ACAGCGGGACTATGATTCT |
| Sequence-based reagent | gDNA | Synthego | | GGACTCTGGAGCGCCTAAAA |
| Software, algorithm | CRIS.py | https://github.com/patrickc01/CRIS.py; *Connelly and Pruett-Miller, 2019* | | CRISPR-editing analysis software |
| Software, algorithm | Sequest (version 28 revision 13) | Mark P Jedrychowski, et al. | | Database search algorithm for mass spectrometry-based protein detection |
| Software, algorithm | JUMPm | St Jude | | Metabolomics data analysis software |
| Software, algorithm | MoNA | https://mona.fiehnlab.ucdavis.edu/ | | MS/MS library, used for metabolomics |
| Software, algorithm | Human Metabolome Database (HMDB) | https://hmdb.ca/ | | MS/MS library, used for metabolomics |
| Software, algorithm | mzCloud | https://mzcloud.org | | MS/MS library, used for metabolomics |
| Software, algorithm | Trim-Galore v0.60 | https://github.com/FelixKrueger/TrimGalore; *Krueger, 2023* | | Software used to trim raw reads |
| Software, algorithm | STAR v2.7 | St Jude | | Pipeline used to map RNA reads to human genome, and differential gene expression |
| Software, algorithm | R limma package v3.42.2 | https://bioconductor.org/packages/release/bioc/html/limma.html | | Software used to normalize and transform read counts |
| Software, algorithm | Molecular Signatures Database (MSigDB 6.2) (gsea2 v2.2.3) | https://www.gsea-msigdb.org/gsea/index.jsp | | Software used to perform gene set enrichment analysis (GSEA) |
| Software, algorithm | rMATS v4.1.0 | https://rnaseq-mats.sourceforge.io/rmats4.1.0/download.html | | Software used for RNA alternative splicing analysis using mapped BAM files as input |
| Software, algorithm | R2 | https://hgserver1.amc.nl/cgi-bin/r2/main.cgi?open_page=login | | Portal used to investigate expression of JMJD6, GAC, and KGA in tumor tissues |
| Software, algorithm | Kocak dataset GSE45547 (649 samples) | https://hgserver1.amc.nl/cgi-bin/r2/main.cgi?open_page=login | | Dataset used to investigate expression of JMJD6, GAC, and KGA in tumor tissues |
| Software, algorithm | Fischer dataset GSE120572 (394 samples) | https://hgserver1.amc.nl/cgi-bin/r2/main.cgi?open_page=login | | Dataset used to investigate expression of JMJD6, GAC, and KGA in tumor tissues |
| Software, algorithm | St Jude cloud | https://pecan.stjude.cloud/ | | Portal used to investigate expression of JMJD6, GAC, and KGA in pediatric tumors |
| Software, algorithm | cBioportal | http://cbioportal.org | | Portal used to investigate copy number alterations of JMJD6 and Kaplan-Meier analyses |
| Software, algorithm | DepMap | https://depmap.org/portal/ | | Portal used to investigate metabolite abundance |
| Software, algorithm | STRING program | https://string-db.org | | Software used for network interaction analysis |
| Software, algorithm | Cytoscape | https://cytoscape.org/ | | Software used for presenting network interactions |

| Reagent type (species) or resource | Designation | Source or reference | Identifiers | Additional information |
|---|---|---|---|---|
| Software, algorithm | CRAN | https://CRAN.R-project.org/package=pheatmap | | CNV heatmap generation |
| Software, algorithm | GraphPad Prism v9 | https://www.graphpad.com/ | | Software used for statistical analysis |
| Software, algorithm | ImageJ | https://imagej.net/ij/ | | Software used for colony formation/density |

