## [Editor Report · eLife assessment]

This **important** study reports on key characteristics of MYC-driven cancers: dysregulated pre-mRNA splicing and altered metabolism, with the data being overall **solid**. The manuscript should be of broad interest to cancer biologists due to its therapeutic implications.

---

## [Referee Report · Reviewer #1 (Public Review)]

Summary of Author's Objectives:

The authors aimed to explore JMJD6's role in MYC-driven neuroblastoma, particularly in the interplay between pre-mRNA splicing and cancer metabolism, and to investigate the potential for targeting this pathway.

Strengths:

(1) The study employs a diverse range of experimental techniques, including molecular biology assays, next-generation sequencing, interactome profiling, and metabolic analysis. Moreover, the authors specifically focused on gained chromosome 17q in neuroblastoma, in combination with analyzing cancer dependency genes screened with Crispr/Cas9 library, analyzing the association of gene expression with prognosis of neuroblastoma patients with large clinical cohort. This comprehensive approach strengthens the credibility of the findings. The identification of the link between JMJD6-mediated pre-mRNA splicing and metabolic reprogramming in MYC-driven cancer cells is innovative.

(2) The authors effectively integrate data from multiple sources, such as gene expression analysis, RNA splicing analysis, JMJD6 interactome assay, and metabolic profiling. This holistic approach provides a more complete understanding of JMJD6's role.

(3) The identification of JMJD6 as a potential therapeutic target and its correlation with the response to indisulam have significant clinical implications, addressing an unmet need in cancer treatment.

Weaknesses:

It would be beneficial to explore whether treatment with JMJD6 inhibitors, both in vitro and in vivo, can effectively target the enhanced pre-mRNA splicing of metabolic genes in MYC-driven cancer cells. However, the authors have noted that there are currently no potent and selective JMJD6 inhibitors available.

Appraisal of Achievement and Conclusion Support:

The authors have effectively met their objectives by offering valuable insights into JMJD6's role in MYC-driven neuroblastoma. The results robustly underpin their conclusions about JMJD6's contribution to metabolic reprogramming through alternative splicing and its connection to the therapeutic response to indisulam.

Likely Impact on the Field and Utility of Methods/Data:

The study's findings have the potential to significantly impact the field of cancer research by identifying JMJD6 as a promising therapeutic target for MYC-driven cancers. The methods and data presented in the manuscript offer valuable resources to the research community for further investigations into cancer metabolism and splicing regulation.

Additional Context for Interpretation:

Understanding the complex interplay between cancer metabolism and splicing regulation is crucial for developing effective cancer treatments. This study sheds light on a previously poorly understood aspect of MYC-driven cancers and opens new avenues for targeted therapies. However, the transition from preclinical findings to clinical applications may face challenges, which should be considered in future research and clinical trials.

---

## [Referee Report · Reviewer #2 (Public Review)]

Summary:

Jablonowski and colleagues explored altered pre-mRNA splicing and metabolism in MYC-driven neuroblastoma cell lines. They focused on the role of JMJD6 assessing cellular transformation, for example through interactions with RNA-binding proteins. Moreover, the study examined JMJD6's impact on the splicing of glutaminase (GLS), crucial in neuroblastoma cell metabolism. It also connected JMJD6 to the anti-proliferative effects of indisulam, a compound targeting RBM39 (splicing factor interacting with JMJD6).

Overall, the findings presented by Jablonowski et al. begin to illuminate a cancer-promoting metabolic, and potentially, a protein synthesis suppression program that may be linked to alternative pre-mRNA splicing through the action of JMJD6 - downstream of MYC. This discovery can provide further evidence for considering JMJD6 as a potential therapeutic target for the treatment of MYC-driven cancers.

Strengths:

Alternative Splicing Induced by JMJD6 Knockdown: the study presents evidence for the role of JMJD6 in alternative splicing in neuroblastoma cells. Specifically, the RNA immunoprecipitation experiments demonstrated a significant shift from the GAC to the KGA GLS isoform upon JMJD6 knockdown. Moreover, a significant correlation between JMJD6 levels and GAC/KGA isoform expression was identified in two distinct neuroblastoma cohorts. This suggests a causative link between JMJD6 activity and isoform prevalence.

Physical Interaction of JMJD6 in Neuroblastoma Cells: The paper provides preliminary insight into the physical interactome of JMJD6 in neuroblastoma cells. This offers a potential mechanistic avenue for the observed effects on metabolism and protein synthesis and could be exploited for a deeper investigation into the exact nature, and implications of neuroblastoma-specific JMJD6 protein-protein interactions.

Weaknesses:

There are several areas that would benefit from improvements with regards to the neuroblastoma modelling strategy, lack of in vivo data, and depth of mechanistic investigation. While the need for additional experimental evidence in these areas remains (as highlighted in the initial review), the authors have now acknowledged several relevant limitations and provided a paragraph discussing future experimental work.

---

## [Author Response]

The following is the authors’ response to the original reviews.

We highly thank the editor and reviewers for their time and insightful comments and suggestions. We have made revisions by performing additional experiments and analysis, and clarified the items based on the suggestions.

**Reviewer #1 (Public Review):**
Summary of Author's Objectives:The authors aimed to explore JMJD6's role in MYC-driven neuroblastoma, particularly in the interplay between pre-mRNA splicing and cancer metabolism, and to investigate the potential for targeting this pathway.Strengths:(1) The study employs a diverse range of experimental techniques, including molecular biology assays, next-generation sequencing, interactome profiling, and metabolic analysis. Moreover, the authors specifically focused on gained chromosome 17q in neuroblastoma, in combination with analyzing cancer dependency genes screened with Crispr/Cas9 library, analyzing the association of gene expression with prognosis of neuroblastoma patients with large clinical cohort. This comprehensive approach strengthens the credibility of the findings. The identification of the link between JMJD6-mediated premRNA splicing and metabolic reprogramming in MYC-driven cancer cells is innovative.(2) The authors effectively integrate data from multiple sources, such as gene expression analysis, RNA splicing analysis, JMJD6 interactome assay, and metabolic profiling. This holistic approach provides a more complete understanding of JMJD6's role.(3) The identification of JMJD6 as a potential therapeutic target and its correlation with the response to indisulam have significant clinical implications, addressing an unmet need in cancer treatment.Weaknesses:(1) The manuscript contains complex technical details and terminology that may pose challenges for readers without a deep background in molecular biology and cancer research. Providing simplified explanations or additional context would enhance accessibility.

We have provided simplified explanations for some terminology.

(2) It would be beneficial to explore whether treatment with JMJD6 inhibitors, both in vitro and in vivo, can effectively target the enhanced pre-mRNA splicing of metabolic genes in MYC-driven cancer cells.

Unfortunately, there is no potent and selective JMJD6 inhibitors available.

**Reviewer #3 (Public Review):**
Summary:Jablonowski and colleagues studied key characteristics of MYC-driven cancers: dysregulated pre-mRNA splicing and altered metabolism. This is an important field of study as it remains largely unclear as to how these processes are coordinated in response to malignant transformation and how they are exploitable for future treatments. In the present study, the authors attempt to show that Jumonji Domain Containing 6, Arginine Demethylase And Lysine Hydroxylase (JMJD6) plays a central role in connecting pre-mRNA splicing and metabolism in MYC-driven neuroblastoma. JMJD6 collaborates with the MYC protein in driving cellular transformation by physically interacting with RNA-binding proteins involved in pre-mRNA splicing and protein regulation. In cell line experiments, JMJD6 affected the alternative splicing of two forms of glutaminase (GLS), an essential enzyme in the glutaminolysis process within the central carbon metabolism of neuroblastoma cells. Additionally, the study provides in vitro (and in silico) evidence for JMJD6 being associated with the anti-proliferation effects of a compound called indisulam, which degrades the splicing factor RBM39, known to interact with JMJD6.Overall, the findings presented by Jabolonowski et al. begin to illuminate a cancer-promoting metabolic, and potentially, a protein synthesis suppression program that may be linked to alternative pre-mRNA splicing through the action of JMJD6 - downstream of MYC. This discovery can provide further evidence for considering JMJD6 as a potential therapeutic target for the treatment of MYC-driven cancers.Strengths:Alternative Splicing Induced by JMJD6 Knockdown: the study presents evidence for the role of JMJD6 in alternative splicing in neuroblastoma cells. Specifically, the RNA immunoprecipitation experiments demonstrated a significant shid from the GAC to the KGA GLS isoform upon JMJD6 knockdown. Moreover, a significant correlation between JMJD6 levels and GAC/KGA isoform expression was identified in two distinct neuroblastoma cohorts. This suggests a causative link between JMJD6 activity and isoform prevalence.Physical Interaction of JMJD6 in Neuroblastoma Cells: The paper provides preliminary insight into the physical interactome of JMJD6 in neuroblastoma cells. This offers a potential mechanistic avenue for the observed effects on metabolism and protein synthesis and could be exploited for a deeper investigation into the exact nature, and implications of neuroblastoma-specific JMJD6 protein-protein interactions.Weaknesses:There are several areas that would benefit from improvements with regard to the current data supporting the claims of the paper (i.e., the conclusion presented in Figure 8).Neuroblastoma Modelling Strategy: The study heavily relies on cell lines without incorporating patient derived cells/biomaterials. Using databases to fill gaps in the experimental design can only fortify the observations to a certain extent. A critical oversight is the absence of non-cancerous control cells in many figures, and the rationale for selecting specific cell lines for assays/approaches remains somewhat unclear. A foundational control for such experiments should involve the non-transformed neural crest cell line, which the authors have readily available. Are the observed splicing and metabolic effects of JMJD6 specific to neuroblastoma? Is there a neuroblastoma-specific JMJD6 interactome? Is MYC function essential?In Vivo Modelling: The inclusion of a genetic mouse model combined with an inducible JMJD6 knockdown, would enhance the study by allowing examination of JMJD6's role during both tumor initiation and growth in vivo. For instance, the TH-MYCN mice overexpressing MYCN in neural crest cells, could be a promising choice.Dependence on Colony Formation Assay: The study leans on 2D and semi-quantitative colony formation assays to assess malignant growth. To validate the link between the mechanistic insights discussed (e.g., reduced protein synthesis) and JMJD6-mediated malignant growth as a potential therapeutic target, evidence from in vivo or representative 3D models would be crucial.Data Presentation and Rigor: The presented data is predominantly qualitative and necessitates quantification. For instance, Western blots should be quantified. The RNAseq, metabolism, and pulldown data should be transparently and numerically presented. The figure legends seem elusive and theirlack of transparency (oden with regards to biological repeats, error bars, cell line used etc.) is concerning. Adequate citation and identification of all data sources, including online resources, are imperative. The manuscript would also benefit from a more rigorous depiction and quantification of RNA interference of both stable and transient knockdowns with quantitative validation at mRNA and protein levels.Novelty Concerns: The emphasis on JMJD6 as a novel neuroblastoma target is contingent on the new mechanistic revelations about the JMJD6-centered link between splicing, metabolism, and protein synthesis. Given that JMJD6 has been previously linked to neuroblastoma biology, the rationale (particularly in Figure 1) for concentrating on JMJD6 may stem more from bias rather than data-driven reasoning.Depth of Mechanistic Investigation: Current evidence lacks depth in key areas such as JMJD6-RNA binding. A more thorough approach would involve pinpointing specific JMJD6 binding sites on endogenous RNAs using techniques such as cross-linking and immunoprecipitation, paired with complementary proximity-based methodologies. Regarding the presented metabolism data, diving deeper into metabolic flux via isotope labeling experiments could shed light on dynamic processes like TCA and glutaminolysis. As it stands, the 'pathway cartoon' in Figure 6d appears overly qualitative.

Response: We agree with this reviewer that more in-depth studies are needed to understand the biological functions of JMJD6 in neuroblastoma. We have included one paragraph “limitation of the study” to point out that additional work needs to be done to address the comments from this reviewer.

We have also added details in figure legend to increase rigor.

**Reviewer #1 (Recommendations For The Authors):**
In this study, Jablonowski and colleagues identify the link between JMJD6-mediated pre-mRNA splicing and metabolic reprogramming in cancer cells, with implications for therapeutic response to splicing inhibitors. I have reviewed your manuscript and found it quite promising. However, there are some specific points that require further clarification and additional experiments. Please consider the following comments:Major concerns:(1) Regarding Figure 1d and e: to enhance the robustness of your findings, it would be beneficial to include additional datasets, such as the Kocak-649 dataset. It is important to narrow down the analysis to high-risk patient groups when examining survival rates, specifically to investigate whether the elevated expression of the 114 gene signature correlates with poor survival within this subgroup. Additionally, please consider conducting a more detailed breakdown of the subsets depicted in Fig. 1b to explore the association between their expression levels and patient survival rates.

Response: We have included the Kocak-649 datasets as Supplemental Figure 1. We have further analyzed the 114 gene signature in low-risk and high-risk patients, respectively, as Supplemental Figure 2.

(2) Fig. 2b: Similar to the previous comment, it would strengthen your findings to include survival rate analysis in more datasets, particularly in high-risk patient groups.

Response: We have further analyzed the association of JMJD6 with survival in low-risk and high-risk patients, respectively, as Supplemental Figure 3. Regardless of the risk factors, high expression of JMJD6 was associated with a poor outcome.

(3) In reference to Fig. S1D, please clarify the time point under investigation. It looks like siRNAs were utilized in this study. Ensure consistency between the siRNA # mentioned in the methods section and what is presented in Fig. S1d.

Response: We have clarified the time point under investigation in Fig. S1D (now as Fig. S4D). We have corrected the siRNA# on the method section.

Additionally, it would be beneficial to include data on knockdown efficacy and consider incorporating western blot results, similar to those presented in Fig. 2c.

Response: These experiments were performed as shown in Figure 4C. We assumed the knockdown efficiency was comparable.

Furthermore, I recommend analyzing the RNA-seq data from JMJD6-depleted BE(2)C cells to identify any alterations in the expression of neuronal differentiation signature genes, with the aim of exploring potential associations with changes in cell morphology showed in Fig. S1D.

Response: We have analyzed the data and indeed like this reviewer expected, we do see the upregulation of neuronal differentiation pathways. We have included the data as Fig. S7B.

(4) Fig. 4g: Confirm whether the data is related to GAC, and if so, where is the data for KGA?

Response: We apologize for this. KGA data was missed when we assembled the figure. We have added back as Figure 4H.

(5) In relation to Fig. 4, I suggest conducting experiments to individually silence GAC and KGA, if feasible (for instance, by targeting their 3'-UTRs). This would allow for a more in-depth investigation into whether GAC and KGA play essential roles in NB cell proliferation.

Response: As this reviewer suggested, we have performed the experiments to knock down GAC and KGA in BE2C cells, and we found that both isoforms seemed to be important for cell survival. We have included the data as Figure 5G-I. Additionally, we have also performed RNA-seq to understand the differential functions of GAC and KGA in neuroblastoma cells when they were overexpressed separately. We have included the data as Figure 5E,F, and Supplemental Figure 9.

(6) Fig. 5c: Could this protein synthesis reduction be attributed to an artificial overexpression of JMJD6? It would be interesting to investigate whether the genetic silencing of JMJD6 has an impact on total protein synthesis.

Response: This is a great question but could be very challenging to have a definitive answer. Since cells are not happy with knockdown of JMJD6, we may have a secondary effect resulting from activation of cell death. While we have successfully generated single cell JMJD6 CRISPR KO clones, the cells are not happy either. In the future, we may generate dTAG knockin cell line which will allow us to induce an acute protein degradation, and then we can assess if JMJD6 loss will consequently impact total protein synthesis.

(7) Fig. S7: the authors have shown that knocking down of JMJD6 in NB cells reduced cell proliferation (Fig. 2c-e). Please clarify how you obtained sufficient cells ader CRISPR knockout of JMJD6 clones and whether the cells remained healthy. It would be helpful to provide cell images.

Response: We harvested cells at different time points in Fig 2C-E, and we have added the information in Figure legends. Cells were not happy ader JMJD6 KD or KO. We therefore harvest cells for Western blot at an early time point while stained cells for survival effect at a late time point.

(8) Fig. 7f: Address the paradox where JMJD-knockdown cells grow slower (Fig. 2c-e), but these JMJD-KO4E5 cells grow at a similar rate compared to SKNAS-WT in the DMSO treatment group. Clarify whether this aligns with the results observed with shRNA results shown in Fig. 2c-e.

Response: The JMJD6 KO cells grew much slower than the wild-type cells. In these experiments, we intentionally seeded a lot more cells for JMJD6 KO clone so that we can have a comparable comparison for the cells with DMSO treatment.

Minor concerns:(1) Fig. 2c: Please specify the time point for Fig. 2c to provide a clearer context for readers.

We have added the information.

(2) In Line 204, it is stated that 'Supplementary Table 3,' which describes the 'Correlation of JMJD6 KO and its co-dependency genes,' can actually be found in 'Supplementary Table 4.' Please clarify this discrepancy.

We apologize for this. We probably accidentally uploaded the duplicates. We have uploaded the new table in our revision.

(3) Line 207: The order of figures should be clarified. Fig. 3c should be mentioned before Fig. 3b in the text.

Yes, we did.

(4) In Line 216, it is mentioned that 'Supplementary Table 4,' which describes 'Differentially expressed genes by JMJD6 KD,' can actually be found in 'Supplementary Table 3.' Please provide clarification for this discrepancy.

We have corrected this.

(5) Line 244-247: Please provide clarification of this section to ensure readers can fully understand your point.

We have rephrased the sentence.

(6) Line 1048: Confirm whether Fig. 2c represents siRNA or shRNA, as the label in the graph does not match the figure legends.

Sorry for this. We have corrected.

(7) Line 1161: Provide clarification regarding the use of Image J from k, and in Line 1162, specify the source of Image J from l.

We apologized for the confusion of our description. We meant “Image J” sodware. We have corrected in Figure legend.

**Reviewer #2 (Recommendations For The Authors):**
Suggestions to authors:Line 39 - suggest introducing JMJD6.

Response: We have added the full name of JMJD6.

Line 47 - suggest slightly rephrasing 'metabolic program that is coupled with...'.

We have made a slight change by changing “coupled” to “associate”.

Line 85 - please delete/replace 'exceptional'; proofread for inadequate use of ambiguous wording.

We have changed it as “significant”.

Line 141 - please concisely define 'high risk'.

We have defined it with a citation (line 142-146).

Line 143 - please concisely define 'event free'.

We have defined the event free and overall survival precisely (line 149, 150).

Line 153 - provide an adequate citation for 'cBioportal'.

We have added the citation (line166).

Line 161 - please state the utilized cell lines.

We have referenced to Materials and Methods (line 175).

Line 166 - please note that 'morphological changes' of a cell do not suffice to determine 'stemness', please rephrase.

We agreed and changed it to “regulate cellular differentiation” (line 181).

Line 182 - provide a quantifiable measure for color change and or remove observation from the narrative.

We have removed “indicative of acidic pH change” (line 198).

Line 185 - the statement commencing with 'It is believed...' requires referencing.

We have added references (line 200).

Line 187 - please provide an adequate citation for the 'JoMa1' neural crest-derived cells (J. Maurer and colleagues?).

We have added the reference (line 201).

Line 203 - please provide an adequate citation for 'DepMap'.

There is no citation specifically for DepMap and that’s why we can only provide the DepMap link.

Line 234 - please provide an adequate citation for 'two algorithms'.

We have provided the reference (line 265).

Line 265 - please provide a rationale for the choice of the three tested cell lines.

We have added definition by saying C-MYC overexpressed SKNAS, BE2C and SIMA with MYCN amplification (line 302, 303).

Line 279 - suggest rephrasing 'gaining more ATPs'.

We have removed these words as we do not have direct evidence to show ATP production (line 320).

Line 342 - suggest rephrasing 'are in the only gene signature'.

We have rephrased by saying “lysine demethylase (HDM) genes, including JMJD6, are present in the most significantly enriched gene signature in indisulam-sensitive cells” (line 416-416).

Line 424 - please state the source or all cell lines (commercial provider?).

We have added the source of cell lines.

Lines 438 to 442 - are STR and mycoplasma profiling data adequately presented in the manuscript?

We routinely test STR and mycoplasma for all cell lines cultured in hood in our Department every month.

Lines 520 onwards - is the JMJD6 knockout generation data (e.g., cell viability upon knockout) adequately presented in the manuscript? Why does the study depend on transient transfection of siRNAs for obtaining mechanistic results?

We created stable JMJD6 KO clones by selecting single cell with complete knockout. Cells are not happy ader KO. siRNA knockdown is a method for relatively acute depletion of JMJD6, which is easy and fast, and may be more reliable to assess the direct effect of JMJD6.

Figures: please provide adequate axis-labeling for all graphs (e.g., FIg2 b, and e).

We have added the axis labeling.

Discussion line 370 - what is meant by 'too harsh' - please use unambiguous phrasing to highlight limitations.

We have changed to “stringent”.

Please provide a study limitation paragraph.

We have added one limitation paragraph.

Limitation of the study

Our study focused on the understanding of JMJD6 function in neuroblastoma cell lines. In the future, we will consolidate our study by expanding our models to patient-derived xenograds, organoids, and neuroblastoma genetic models, in comparison with non-cancerous cells. Although we have identified a conserved interactome of JMJD6 in neuroblastoma cells, it remains to be determined whether it is neuroblastoma-specific and essential to MYC-driven cancers. The genome-wide RNA binding by JMJD6 in cancer cells and normal cells coupled with isotope labeling to dissect the metabolic effect of JMJD6 will enhance our understanding of the biological functions of JMJD6, awaiting future studies. Inability to target the enhanced pre-mRNA splicing of metabolic genes in MYC-driven cancer cells by pharmacologic inhibition of JMJD6 is another limitation, due to lack of selective and potent JMJD6 inhibitors.

Additional editing and proof-reading of the manuscript's narrative, figures, legends, and methods is highly recommended.

We have gone through the whole MS to have proof-reading.